behaviour

compression, communication, Zipf, Menzerath, language

**Author for correspondence:**
Catherine Hobaiter
e-mail: clh42@st-andrews.ac.uk

†These authors contributed equally to this work.

# Variable expression of linguistic laws in ape gesture: a case study from chimpanzee sexual solicitation

Alexandra Safryghin[1], Catharine Cross[1], Brittany Fallon[1], Raphaela Heesen[3], Ramon Ferrer-i-Cancho[2,†] and Catherine Hobaiter[1,4,†]

[1]School of Psychology and Neuroscience, University of St Andrews, St Andrews, Fife, UK
[2]Complexity and Quantitative Linguistics Laboratory, Laboratory for Relational Algorithmics, Complexity, and Learning Research Group, Departament de Ciències de la Computació, Universitat Politècnica de Catalunya, 08034 Barcelona, Catalonia, Spain
[3]Department of Psychology, Durham University, Durham, UK
[4]Budongo Conservation Field Station, Masindi, Uganda

CC, 0000-0001-8110-8408; RH, 0000-0002-8730-1660; CH, 0000-0002-3893-0524

Two language laws have been identified as consistent patterns shaping animal behaviour, both acting on the organizational level of communicative systems. Zipf's law of brevity describes a negative relationship between behavioural length and frequency. Menzerath's law defines a negative correlation between the number of behaviours in a sequence and average length of the behaviour composing it. Both laws have been linked with the information-theoretic principle of compression, which tends to minimize code length. We investigated their presence in a case study of male chimpanzee sexual solicitation gesture. We failed to find evidence supporting Zipf's law of brevity, but solicitation gestures followed Menzerath's law: longer sequences had shorter average gesture duration. Our results extend previous findings suggesting gesturing may be limited by individual energetic constraints. However, such patterns may only emerge in sufficiently large datasets. Chimpanzee gestural repertoires do not appear to manifest a consistent principle of compression previously described in many other close-range systems of communication. Importantly, the same signallers and signals were previously shown to adhere to these laws in subsets of the repertoire when used in play; highlighting

that, in addition to selection on the signal repertoire, ape gestural expression appears shaped by factors in the immediate socio-ecological context.

## 1. Introduction

Over the past 100 years, important statistical regularities have been described across human languages and in other communicative systems such as genomes, proteins and animal vocal and gestural communication [1–11]. These regularities are hypothesized to be manifestations of the information-theoretic principle of compression [9,12]. Compression is a particular case of the principle of least effort [13]—a principle that promotes the outcome that requires the least amount of energy to produce or achieve—and thereby promotes coding efficiency [14]. In communication, compression is expressed as pressure towards reducing the energy needed to compose a code but limited by the need to retain the critical information in the transmission [15,16].

Among the statistical patterns predicted by compression at different levels of organization, Zipf's law of brevity and Menzerath's law have been at the centre of recent attention in studies of human and non-human communication. Zipf's law of brevity is the tendency for more frequent words to be shorter in length [13,17], and is generalized as the tendency for more frequent elements of many kinds (e.g. syllables, words, calls) to be shorter or smaller [14]—with similar patterns found at different levels of analysis, for example in speech at the level of words [17] syllables [18], and phonemes [4]. As well as being found in human spoken, signed and written languages [2–4,8,10], Zipf's law of brevity has been identified in the short-range communication of diverse taxa: dolphins [16], bats [19], penguins [20], hyraxes [21] and various primates (macaques: [22]; marmosets: [23]; gibbons: [24]; *Indri indri*: [25]), as well as in genomes [7].

At the level of constructs, Menzerath's law states that '*the greater the whole, the smaller its constituents*' [6,26,27]; for example longer sentences have words of shorter average length, and words with more syllables contain syllables of shorter length. Menzerath's law (and its mathematical expression known as the Menzerath-Altman's law) has been identified in human spoken and signed languages [26,28], genomes [29,30], music [31], and in the communication of dolphins [16], penguins [20] and primates (geladas: [32]; chimpanzees: [33,34]; gibbons: [24,35]; gorillas: [36]; *Indri indri*: [25]). While many studies focused on vocal communication, several have now explored these statistical regularities in gestural and signed domains. For example, the use of Swedish Sign Language in (semi-)spontaneous conversation was found to follow a pattern of more frequently used signs being shorter in duration [3]. Zipf's law of brevity was also found in fingerspelling, with a negative relationship between mean fingerspelled sign duration and frequency [3]. Similarly, Czech sign language was found to follow Menzerath's law [28]. Work in non-human gesture has, to date, been more focused on context-specific signal usage, for example Zipf's law of brevity was found in the surface behaviour of dolphins (such as tail-slapping; [37]) but not in the overall repertoire of play gestures of chimpanzees, where it was only present in subsets, although these gestures did follow Menzerath's law [34].

Chimpanzee gestural communication represents a powerful non-human model in which to explore compression and language laws. Apes have large repertoires of over 70 distinct gesture types [38]; as compared to vocal communication, gestural repertoires are larger and are more flexibly deployed, with individual gesture types used to achieve multiple goals [39–42]. Gestures are also used intentionally, i.e. to reach social goals by influencing the receivers' behaviour or understanding [41,43–45]), and flexibly across contexts [40–42]. Nevertheless, Heesen *et al.*'s [34] results support an increasingly diverse range of findings that show variation in the extent and expression of language laws, suggesting that while they appear statistically universal there is room for exceptions and/or variation in patterning at different levels of the communicative construct [9].

Although a lack of evidence supporting Zipf's law of brevity has been previously reported (e.g. European heraldry: [46]; computer-based neural networks: [47]), these remain rare exceptions, and in non-human animal communication have typically only been reported in long-distance vocal communication (e.g. gibbon song: [35]; bats: [19]; although cf. female hyrax calls: [21]) where the impact of distance on signal transmission fidelity may have a particularly strong effect on the costs of compression [9,14,32]. Thus, at present, the repertoire-level absence of Zipf's law of brevity in chimpanzee gesture remains a conundrum.

One explanation for a repertoire-level absence of Zipf's law of brevity—as seen in some long-distance signals—is that the context in which signals are produced may impact the emergence and expression of

these patterns. Specifically, in the case of chimpanzee gestures, the absence of a pattern resembling Zipf's law of brevity may result from the use of gestures produced during play. Expressions of linguistic laws in biological systems reflect pressures that shape efficient energy expenditure [9]. Play is produced when there is an excess of time and energy [48–50], thus, the energetic need to reduce signal effort through increased compression may be limited. As a result, it remains unclear whether the failure of Zipf's law of brevity in chimpanzee gesture was owing to the use of gestures from within play, or whether it reflects a system-wide characteristic.

In both signed languages and human gesturing, distinctions are made between different components of their production. First there is the *preparation* of the signal, then the *action stroke* which represents the movement that defines the gesture as of a particular type; an individual can then choose to further *hold* the stroke or repeat it, until they decide to stop gesturing and return the limb to rest during *recovery* from the gestural action [51]. For example: in a reach gesture this would correspond to the movement of the hand into position (*preparation*), the extension of the arm and hand towards the recipient (*action stroke*), the (optional) maintenance of the extension (*hold*), and finally the return of the hand and arm to a resting state (*recovery*). All four of these phases require some energetic investment to produce, but there may be variation across them, and aspects such as preparation and recovery may be nearly, or entirely, absent where several gestures are strung together. In some gesture types, their production does not include a *hold* phase (e.g. hit, jump, throw object); we term these *fixed* duration gestures, as the duration of their expression is relatively constrained across instances of production. Other gesture types can include a *hold* phase (for example: reach, object shake, swing) which may or may not be present, and, where present, may vary substantially in length; we term these *loose* duration gestures. There may be differences in the emergence of Zipf's and Menzerath's laws regarding the different components of gesture production. Menzerath's law acts from a proximate perspective on the building of communicative sequences in a specific communicative instance: for example, gestures produced in longer sequences may be shortened by variation the duration of components such as the shortening of the *hold* phase in loose gesture types. By contrast, Zipf's law acts on gesture types across instances of use—and as such may be less sensitive to the immediate context of production.

Another possible explanation for the variation in the emergency of compression in ape gesture is that the ability to detect linguistics laws, particularly where they are only subtly expressed, appears to require powerful datasets. The exploration of statistical patterns in human languages often employs corpora containing millions of data points (e.g. [52]). By contrast, in ape gesture, as in many studies of non-human communication, datasets are substantially smaller (in the thousands). In chimpanzee play, the large repertoire expressed limits the frequency with which particular gesture types are represented.

We address this open question in a case study of chimpanzee gestural communication in sexual solicitation. While gesture is relatively under-studied in this area, sexual solicitations have been contrasted with early descriptions of gesture from studies of captive ape play, as an example of gesture in a relatively more evolutionarily or biologically 'relevant' context for communication (in terms of associated risks and/or impact on reproduction) ([53]; cf. [40]). Chimpanzees, particularly male chimpanzees, employ prolific use of individual gestures and gesture sequences in sexual solicitations. As solicitations are often vigorous, chimpanzees incorporate regular use of gesture types that include both visual and audible information [53,54]. While a range of gesture types are employed, these are typically a smaller sub-set of the available repertoire—cf. play where the majority of gesture types are deployed. Successful gestures can lead directly to sexual behaviour, such as inspection or copulation, as well as to a consortship, in which the female follows the male away from other individuals in the group so that he maintains exclusive sexual access [55]. Both direct solicitation and consortship and are key strategies for individual fitness [55,56], and as such behaviour associated with them is probably subject to strong selective pressures. The energetic costs of lactation mean that adult female chimpanzees typically concieve only once every 4–5 years [57,58]. So while there are typically 60–80 individuals in a group, the operational sex ratio of available females in oestrus may be very small, and males show substantial variation in reproductive success [55,59]. Although highly important, the performance of sexual solicitations may come with significant costs: besides the energetic expenditure in producing these signals, there is a risk of potentially aggressive competition both from other males in their own community [55,60] as well as potentially lethal attacks from males in neighbouring groups [61]. For example, during consortships individuals may travel to the boundaries of their home area, increasing the risk of encounters with neighbouring individuals. Thus, there are substantial advantages to avoiding potential eavesdroppers within, and particularly outside of, one's community [62]. Therefore, on one hand individuals benefit from producing conspicuous energetic signals to attract females, often having to insist to secure mating; on

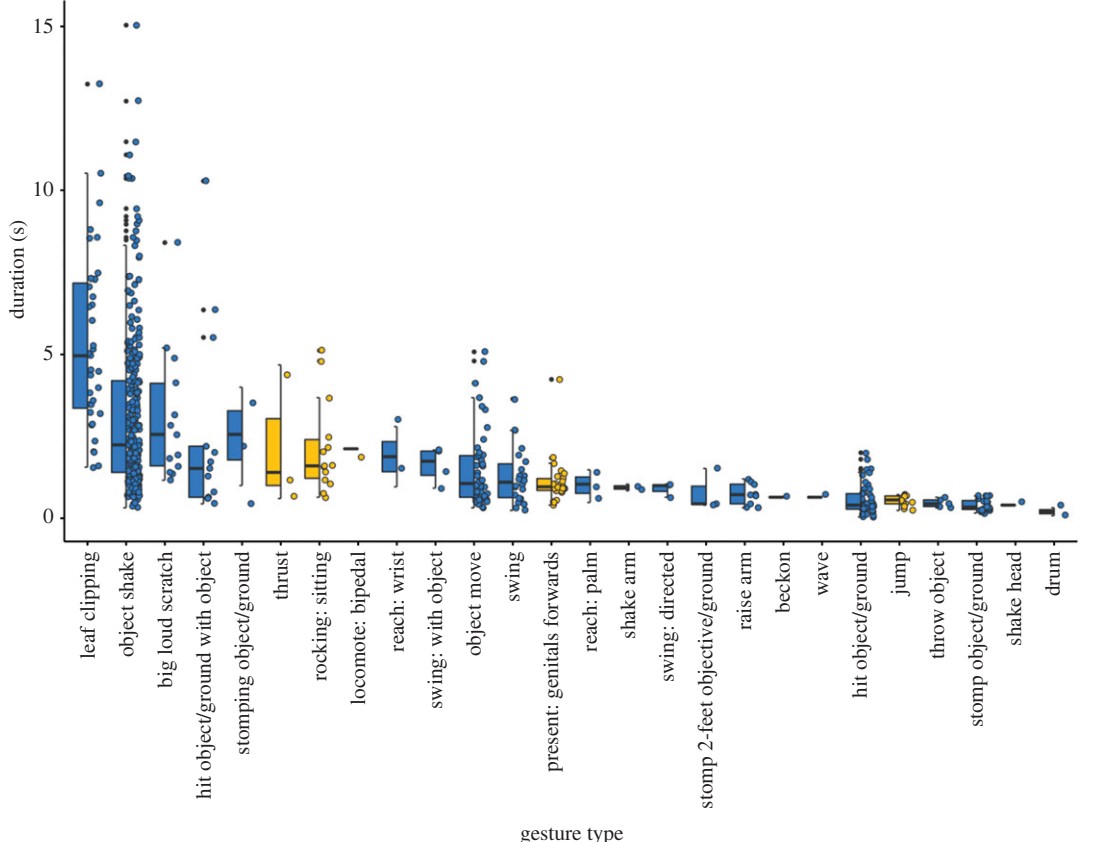

**Figure 1.** Distribution of gestural instances across the 26 gesture types detected with relative gesture duration. Boxplots represent the median (black bar), the interquartile range (boxes), maximum and minimum values excluding outliers (whiskers) and outliers (black dots). Points represent individual gestures. Whole-body gestures are indicated in yellow, manual gestures in blue.

the other, the production of highly conspicuous signals should be compressed to reduce the risks associated with competition from both within and outside the group.

To assess compression in the sexual solicitation gestures of wild male chimpanzees, we tested for patterns predicted by Zipf's law of brevity and Menzerath's law, both at the level of single gesture types and gesture sequences, respectively. To investigate Zipf's law of brevity and Menzerath's law we fitted two generalized linear mixed models. The first model explored the presence of Zipf's law assigning gesture duration as the response variable, proportion of gestures within the dataset and category of gesture (manual versus whole body) as fixed factors, and signaller's identity (ID), sequence ID and gesture type as random factors. The second model tested for Menzerath's law and had gesture duration as response variable, sequence size as a fixed factor, and proportion of whole-body gestures in the sequence (PWB), signaller ID, sequence ID and gesture type as random factors. We included information on the category of the gesture to allow for comparisons with human studies, in which gestures are mostly manual. We provide matched models that describe the patterns of expression both across (i) all males in our data, (ii) for a single prolific individual and (iii) for the remaining individuals. In doing so, we provide an initial assessment of the distribution of our findings across male chimpanzee gesturing in this context and provide an expanded assessment of compression in ape gestural communication.

## 2. Results

We measured $n = 560$ sexual solicitation gestures from 173 videos of 16 wild, habituated male East African chimpanzees (*Pan troglodytes schweinfurthii*) gesturing to 26 females. Within the 560 gestural instances (from now *tokens*), we identified 26 gesture types: 21 manual gestures and five whole-body gestures (figure 1; for definitions for full repertoire definitions see the electronic supplementary material, S1 and table S1) performed by 16 male chimpanzees aged 10–42 years old. On average, each

**Table 1.** Number of sequences composed of the same or different gesture types, listed according to sequence length. (Sequence length is defined by the number of gesture tokens present in the sequence.)

| sequence length | the same type | different types | total number of sequences |
|---|---|---|---|
| 1 | NA | NA | 244 |
| 2 | 24 | 58 | 82 |
| 3 | 1 | 20 | 21 |
| 4 | 0 | 3 | 3 |
| 5 | 1 | 6 | 7 |
| 6 | 0 | 2 | 2 |
| total | 26 | 89 | |

individual produced a median of 11.5 ± 70.7 gesture tokens (range 2–290). One male, Duane, was particularly prolific ($n = 290$ gesture tokens; other males 2–76). To provide context as to what extent our findings are generalizable, we provide matched analyses using both the full dataset and the dataset limited to Duane only. An analysis of the data excluding Duane is available in the electronic supplementary material.

Gesture token duration was measured via analysis of video data with a minimum unit of 0.04 s (one frame). Duration ranged from 0.04–15.04 s (median: 1.56 ± 2.35 s). If consecutive gesture tokens were performed with less than 1 s in between them, they were considered to form a sequence [34,63]. We detected a total of 377 sequences, with each male performing a median of 8 ± 44.54 sequences (range 1–181 sequences). Sequence length ranged from 1 to 6 tokens (table 1). For analyses of Menzerath's law we excluded 18 sequences for which we were unable to identify the duration of all the consecutive gesture tokens performed, resulting in the analysis of 359 sequences, containing a total of 530 gesture tokens. There were 244 sequences composed of a single token, the remaining 115 sequences had length $n > 1$. Of the 115 sequences analysed that were composed of two or more gesture tokens; 26 (23%) were formed by the repetition of the same gesture type, whereas the remaining 89 (77%) included more than one gesture type (table 1).

## 2.1. Do chimpanzee sexual solicitation gestures follow Zipf's law of brevity?

To test for Zipf's law we ran a Bayesian generalized linear model (Zipf-model), with the log of gesture duration as the response variable and the proportion of gesture type within the dataset as a fixed factor (see the electronic supplementary material, S2 for further detail). The gesture duration data was log-transformed following an analysis of data distribution. We included category of gesture as a control, and signaller ID, sequence ID and gesture type as random factors. The Zipf-model fitted the data better than a null model that did not include the proportion of gesture type as a fixed effect (leave-one-out (LOO) difference and s.d. = −0.7 ± 0.3). For Zipf-model effects bulk effective sample size (ESS) and tail ESS were > 100 and $\widehat{R} = 1$. However, the proportion of gesture type did not have a substantial effect on the duration of gestures (electronic supplementary material, S3 and table S5; $b = 0.90$, s.d. = 1.26, 95% credible intervals (CI) [−1.25, 3.81]; figure 2a). When testing the subset of data containing only the gestures produced by Duane, the full model and null model testing for Zipf's law showed similar fit (LOO difference: −0.1 ± 0.7; electronic supplementary material, S3 and table S6; figure 2b). Similarly, in the same analysis on data from all individuals except Duane, the full model was no different from the null model (LOO difference: −0.5 ± 0.5; electronic supplementary material, S3 and table S7).

## 2.2. Do chimpanzee sexual solicitation gesture sequences follow Menzerath's law?

To test for Menzerath's law we ran a second Bayesian model (Menzerath-model) with the log of the gesture duration as response variable, the sequence size as fixed factor, the PWB as a control, and the signaller ID and sequence ID as random factors. The Menzerath-model fitted the data better than the null model (LOO difference: −7.7 ± 4.1). All predictors had bulk ESS and tail ESS > 100 as well as $\hat{R}$ values = 1. Sequence size had a substantial negative effect on gesture duration within sequence (electronic supplementary material, S3 and table S8; $b = −0.18$, s.d. = 0.04, 95% CI [−0.26, −0.11]; figure 3a). Similar results were found when

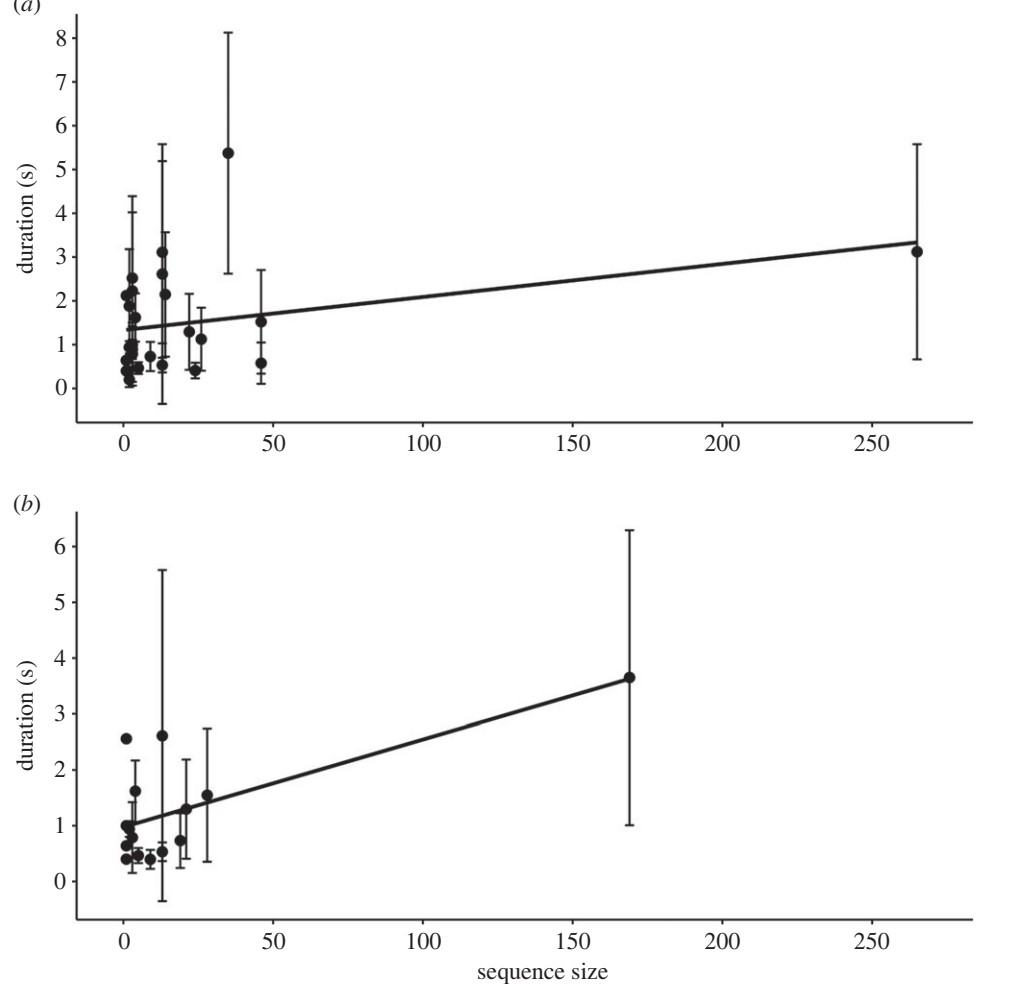

**Figure 2.** Relationship between frequency of occurrence and gesture duration for the full dataset (*a*) and Duane only data (*b*). Points represent the mean duration of each gesture type, with error bars showing the standard deviation from the mean. Black line indicates regression slope.

running the same Menzerath-model but limited to gestures produced by Duane: the full model fitted the data better than the null (LOO difference: $-13.5 \pm 4.5$), all predictors had bulk ESS and tail ESS > 100, $\hat{R} = 1$ and sequence size had a substantial negative effect on gesture duration (electronic supplementary material, S3 and S9; figure 3*b*; $b = -0.23$, s.d. = 0.04, 95% CI [$-0.31$, $-0.15$]). By contrast, where Duane's data were excluded, the full model was similar to the null model, suggesting no clear pattern consistent with Menzerath's law (LOO difference: $-0.3 \pm 0.8$; electronic supplementary material, S3 and table S10). Visual inspection of the data plotted per individual suggests that detection of a pattern consistent with Menzerath's law may be impacted by sample size (electronic supplementary material, S4).

We note that the sample size of sequences of four tokens or longer is smaller than those of one to three tokens (table 1), which may have contributed to the apparent tailing off of a clear relationship in figure 3*a*, *b*. In addition, longer sequences were formed of (i) a mix of *loose* and *fixed* duration gestures, or (ii) only *loose* duration gestures (see the electronic supplementary material, S5 and figures S5 and S6). Thus, the emergence of Menzerath's law could not be explained by a shift in preference from *fixed* to *loose* gestures with increasing sequence length.

## 3. Discussion

Chimpanzee sexual solicitation gestures did not follow Zipf's law of brevity: the frequency of gesture type within the dataset did not predict gesture duration in any of our samples. However, sequences of chimpanzee solicitation gestures did follow Menzerath's law: longer sequences of gestures were made up of gestures of shorter average length. Our dataset was limited both by its relatively small size (cf.

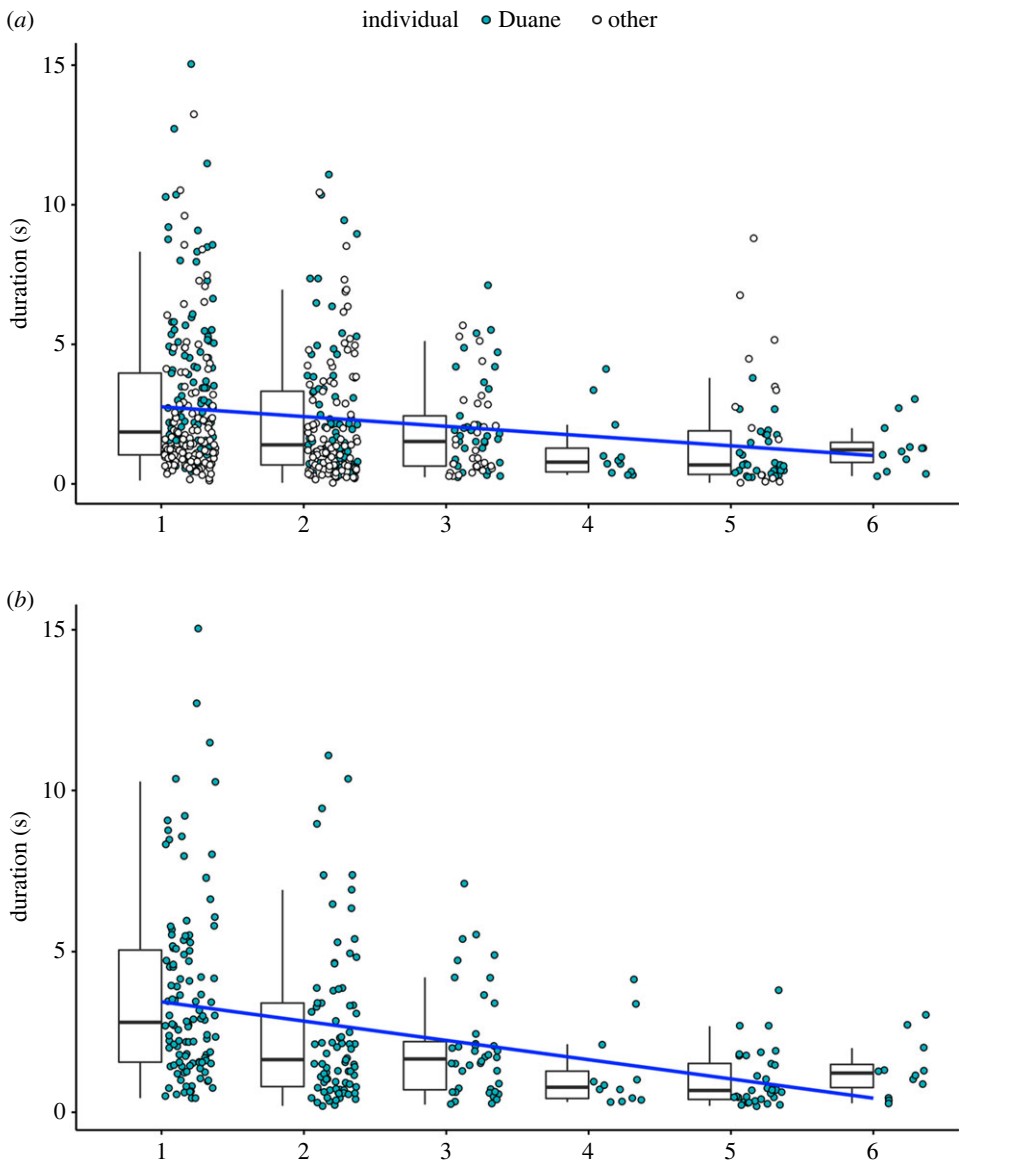

**Figure 3.** Relationship between sequence size and gesture duration for the full dataset (*a*) and Duane only data (*b*). Boxplots represent the median (black bar), the interquartile range (boxes), and maximum and minimum values excluding outliers (whiskers). Points represent individual gesture tokens, ordered by the length of the sequence they were performed in. Gestural tokens belonging to the individual Duane are indicated in light blue. White circles indicate gesture tokens belonging to all other individuals. Blue line indicates regression slope.

[34] on chimpanzee play gestures) and in its bias towards a single highly prolific individual (Duane). As a result, we consider it a case-study; however, the pattern was present in both Duane's data and in the full dataset, as well as in a range of alternative analyses (electronic supplementary material, S6). In the reduced dataset excluding Duane we did not find a pattern consistent with Menzerath's law; however, detection of the pattern may have been limited by the small sample size available in the remaining dataset.

These results represent a further absence of evidence in support of Zipf's law of brevity in great ape gestural communication [34] and support the wider finding that—unlike most other close-range systems of communication described to date—the expression of pressure for compression and efficiency may be variably expressed in ape gesture [3,9,14]. It particularly highlights that compression does not act on communicative systems uniformly: 20 of the 26 gesture types described here as used in sexual solicitations overlapped with those used in play [34]. Data were collected from the same community over the same period, and although both studies provided a null result when analysing the full

gestural repertoire, Zipf's law was found in subsets of the play gestures but not in the gestures when used in sexual solicitations. Moreover, when running traditional correlation analyses in which features such as signaller identity, or gesture type could not be controlled for, we found a tendency for an opposite Zipf's law pattern—particularly in manual gestures (electronic supplementary material, S6). Visual inspection of figure 3 shows the substantial variation in the duration of gestures across instances of communication, as well as an apparent decrease in a clear relationship between gesture duration and sequence size where sample size was small (such as for longer sequences). Together these findings suggest that the expression of these laws is nuanced by aspects of the communicative landscape in which they are deployed, and that large samples may be needed to detect sometimes subtle relationships. Future work could specifically explore variation in the detection of these patterns at different sample sizes, for example by randomized subsetting of sufficiently large datasets. As Semple *et al.* [9] suggest, apparent 'failures' may be of substantial assistance in exploring the boundaries of the theoretical framework of these laws, helping to define the characteristics that shape both their emergence and variation in their expression.

By contrast to vocal communication across primate species, in chimpanzee sexual solicitations 'inefficiency' in signalling effort by the signaller appears to be at times slightly favoured. However, these gestures appear to remain effective in terms of achieving the signaller's goal of successful communication in a context vital for reproductive success. Given the long inter-birth intervals and active mate guarding [64], chimpanzee paternity is often heavily biased towards higher-ranking individuals [59]. With so few opportunities to mate, sexual solicitations may represent one of the most evolutionarily important contexts in which chimpanzee gestures are produced. Where the costs of signal failure are high, there is a pressure against compression and towards redundancy, as in chimpanzees' use of gesture-vocal signal combinations in agonistic social interactions [62]. While there are examples of vocal communication systems used in biologically 'relevant' contexts that adhere to Zipf's brevity law [20], the benefits of successful communication to individual fitness in chimpanzee solicitation appear to outweigh the energetic costs associated with the production of a vigorous and conspicuous signal. Nevertheless, given that we see a relatively consistent expression of Menzerath's law across gesture use in sexual solicitation as in play, even the production of these prolonged and conspicuous signals appear to remain constrained by physiological mechanisms of gestural production. As for primate vocal communication [32,33], where breathing constraints and energetic demands of vocal production were considered drivers for the emergence of Menzerath's law patterns, increased muscular activity related to the production of sequences of gestures [65] could be a general limit on energetic investment. As a result, Menzerath's law appears to emerge across communicative contexts.

There are a number of potential reasons for why language laws appear variable in their expression within ape gesture. For example, we might be considering the wrong unit of analysis. In human speech, sign, and gesture—as in other communication systems—it is possible to consider the production of a 'unit' of communication at different levels. For example, while Zipf's law is clearly expressed in the duration of male rock hyrax vocalizations, it is not the case for female vocalizations where Zipf's law of brevity emerges when analysing call amplitude rather than duration [21]. Conversely, in Börstell *et al.* [3] research on Swedish Sign Language, Zipf's law of brevity seems to hold across sign categorization, fingerspelling and compounding. Interestingly, this study excluded the hold phase of a sign, limiting their analysis only to the more active stroke phase. The production of intentional gestures in apes are shaped not only by the signaller but by the interaction between signaller and recipient [38,66]. As a result, the duration of hold or repetition phase may be shaped by the immediate context of the specific interaction—for example, in waiting for a response by the recipient it may vary between being absent and very prolonged. By contrast, the *action stroke* of a sign or gesture is always present and represents the need to convey information in that gesture, i.e. to discriminate it from other gesture actions. In Swedish Sign Language a prolonged and repetitive feedback sign and prolonged turn-taking signs were the only two cases that diverged from the general Zipf's pattern, as they were both long in duration as well as being highly frequent [3]. Zipf's law acts on a signal 'type' in an individual's or species' repertoire—and it may be of interest to compare its expression across areas of gesture production that are more consistently produced across usage, such as the action stroke.

Research to date has typically focused on signal compression at the level of the communication system, but communication happens *in situ*. Signallers probably respond to pressures on signalling efficiency more broadly: an intense but time-limited investment in clear signalling may be more energetically efficient than the need to travel with a female for extended periods following a failed signal. A similar solicitation with a different audience may need to be produced rapidly and inconspicuously, as the detection of this activity by other males could be fatal [60]. In a recent human

study, pressures towards efficiency and accuracy were both required for Zipf's law of brevity to emerge in experimental communicative tasks between two participants [67]. Conversely, when participants were required to produce solely time-efficient versus solely accurate communicative signals no pattern emerged. The sexual solicitation context tested in our study may mirror the pattern seen in the time-efficient paradigm in the human study. In play, where urgency and time-efficiency may be less relevant, the same signals used by the same chimpanzees did show compression. While many vocalizations are relatively fixed [68,69], gestural flexibility (in goal and context—[39–42]) allows us to explore how compression acts within both specific instances of communication as well as on whole communication systems. To do so will require large longitudinal datasets in which it is possible to test both between-individual variation and within-individual variation across different gesture types and sequence lengths. Similarly, there remains substantial work needed to explore variation across different socio-ecological contexts of gesture use, for example in the social relationship between the signaller and recipient [66]. The use of redundancy within specific subsets of gestural repertoire, or within specific contexts of gesture demonstrates both the importance of compression in communicative systems in general, but also the flexibility present in each specific usage. In doing so, it highlights the importance of exploring the impact of individual and socio-ecological factors within wider patterns of compression in biological systems in evolutionary salient scenarios.

# 4. Methods

We measured $n = 560$ male-to-female sexual solicitation gestures from 173 videos recorded within a long-term study of chimpanzee gestural communication depicting 16 wild, habituated East African chimpanzees (*Pan troglodytes schweinfurthii*) from the Sonso community of the Budongo Forest Reserve in Uganda (1°35′ and 1°55′ N and 31°08′ and 31°42′ E), collected between December 2007 and February 2014. Observations were made between 7.30 and 16.300 with recording of gestures following a focal behaviour sampling approach [70]. Here, all social interactions were judged to have the potential for gesture, in practice any situation in which two chimpanzees were in proximity and not involved in solitary activities, were targeted. Where several potential opportunities to record co-occurred, preference was given to individuals from whom fewer data had been collected (with a running record of data collection maintained to facilitate these decisions).

During October 2007 to August 2009 a Sony Handycam (DCR-HC-55) was used. Here video was recorded on MiniDV tape. The challenges of filming wild chimpanzees in a visually dense rainforest environment meant that, at times, the start of gestural sequences was not captured on video. Where this occurred, it was dictated onto the end of the video and these sequences were not included in analysis. Similarly, sequences in which part of the sequence was obscured, for example where a chimpanzee moves through dense undergrowth, were also discarded. After 2009 video data were collected using Panasonic camcorders (V770, HC-VXF1) which have a 3-second pre-record feature that improves the ability to capture the onset of behaviour; however, the same procedure was used and any sequences where the onset of gesturing was not clearly captured continued to be discarded.

## 4.1. Sexual solicitation gestures

Sexual solicitation gestures were defined as those gestures given by a male towards a female with the goal of achieving sex, usually accompanied by the male having an erection and the female being in oestrus [41,53]. We included solicitations in the context of sexual consortship; here a male gestures in order to escort a female away from the group to maintain exclusive sexual access, which can occur prior to the peak of the female oestrus [55]. We restricted our analyses to male to female sexual solicitation, as female to male sexual solicitation attempts rarely involved sequences of gestures in this population. We further restricted analysis to solicitations by male individuals of at least 8 years old, as this is the minimum age of siring recorded in this community, limiting our signals to those on which there is more direct selective pressure.

## 4.2. Defining gesture types and tokens

In quantitative linguistics, word *types* are used to assess Zipf's law of brevity, whereas *tokens* are used to assess patterns conforming to Menzerath's law. To distinguish the two, consider the question:

Which witch was which?

The question is composed of 4 *tokens* (overall word count), and three different word *types*, (which, witch, was). Gesture *types* (see the electronic supplementary material, table S4 for a detailed repertoire description) were categorized according to the similarity of the gesture movement, which could be used either as a single instance or in a sequence; and each gestural instance represented an individual *token*.

Great apes deploy gestural sequences in two distinct forms [63]: one is the addition of further gestures following response waiting and is typically described as persistence (which may include elaboration). The second is the production of gestures in a 'rapid sequence'—here gestures are produced with less than 1 s between consecutive gesture tokens, and do not meet behavioural criteria for response-waiting occurring within a sequence (although it may occur at the end of it). As the expression of Menzerath's law is typically considered at the level of a unique sequence, rather than one generated through the addition of gestures in response to earlier failure, we limit our analyses here to rapid sequences only. Sequence length was quantified as the number of gesture tokens produced with less than 1 s between two consecutive gesture tokens; single gestures were coded as sequences of length one [34,63].

## 4.3. Gesture duration

Gesture duration was calculated using MPEG streamclip (v. 1.9.3beta). We measured gesture duration in frames, each lasting 0.04 s. Gestural 'units'—like many other signals—can be considered at different levels of analysis, for example: a word is composed of syllables, and syllables of phonemes. Gestures have been described as composed of preparation, action stroke, hold or repetition and recovery phase [51]. Here we follow previous work in [34] in defining the start of a gesture token as the initial movement of a part of the body required to produce the gesture. The end of a gesture token corresponded to (i) the cessation of the body movement related to gesture production, (ii) a change in body positioning if the gesture relied on body alignment, or (iii) the point at which the goal was fulfilled, and any further movement represented effective action (for example, locomotion or copulation). Where the expression of a gesture token did not include a full recovery (in which the body part involved is returned to a resting state), the end of a token was discriminated from subsequent tokens through (i) a change in gesture action, e.g. from a reach to a shake, (ii) a change in orientation or rhythm of a gesture action, hold or repetition, e.g. the rhythm or direction of an object shake is broken or changed [62].

## 4.4. Intra-observer reliability

Video-based coding offers the opportunity to conduct reliability measures. Intra-observer reliability was tested by randomizing the order of the videos and re-coding the duration of the gestures of every ninth clip, for a total of 75 gestures from 23 clips. We performed an intraclass correlation coefficient (ICC) test—class 3 with $n = 1$ rater [71]—which revealed very high agreement on gesture duration measurements (ICC = 0.995, $p < 0.001$). Unfortunately, an additional step of inter-observer reliability was not possible owing to the loss of the file that linked the original dataset to the videos from which data were extracted.

## 4.5. Statistical analysis

All data were analysed using R v. 4.0.0 and RStudio v. 1.2.5042 [72,73]. We fitted Bayesian generalized linear multivariate multilevel models using the 'brm' function from the 'brms' package [74] with minimally informative priors, 2000 iterations and three chains.

We ran a first model testing Zipf's law of brevity (Zipf-model), containing gesture token duration (s) as the response variable, the proportion of occurrences of a particular gesture type in the dataset (proportion) as a fixed effect, and gesture category (manual versus whole-body) as a control. We included signaller ID, sequence ID and gesture type as random effects. We include category as a variable here to allow for more direct comparison with previous work, which often excludes or differentiates non-manual signals, either in great ape gesture [34,75] or in signed languages and fingerspelling (e.g. [3]).

We tested Menzerath's law by running a second model (Menzerath-model) containing gesture token duration (s) as the response variable, sequence size (number of gesture tokens within the sequence) as a fixed factor, and the PWB as a control. We modelled signaller ID and sequence ID as random factors.

It was highlighted during the review process that the emergence of Menzerath's law may be an artifice created by the selection of *fixed*, as opposed to *loose*, duration gesture types when producing

longer sequences. To address this hypothesis, we produced histograms depicting the distribution of *loose* and *fixed* duration gestures within sequences at each sequence size. The majority of gesture types ($n = 20$ of total 26), and of gesture tokens ($n = 456$ of total 560) were of the *loose* gesture form, thus there were very few gesture sequences formed only of *fixed* gesture types. However, we further visually assessed the distributions of *fixed* gestures in sequences formed of only *fixed* gestures.

As our data may be particularly influenced by a single prolific individual (Duane) who contributed around half of the data, we assess the generalizability of our findings by replicating analyses conducted on the full dataset on a subset of the data containing only gestures by Duane as well as on a subset containing all but the prolific individual Duane. For the models testing Duane's data, signaller ID was removed from the random factors as it was no longer relevant (with the inclusion of only one individual). In order to avoid inflation of the dataset we include date as a random factor; which also allows us to avoid biasing the analysis towards particularly prolific days and control for within-individual consistency.

We ran full-null model comparisons using the LOO information criterion [76] 'loo_compare' function from the 'stan' package (v. 2.21.5; [77]) where Zipf's null model contained only the control variable 'category' and the random effects, whereas Menzerath's null model contained only the control variable PWB and the random effects. Prior to the Bayesian analysis we assessed data distribution using the 'fitdistr' package (v. 1.0–14; [78]). Following data inspection, we log-transformed gesture duration and average sequence duration as data from the response variable strongly skewed towards zero (for data inspection see the electronic supplementary material, S2).

Finally, previous work has frequently employed correlation and compression tests, which looks at whether the expected mean code length observed in the dataset is significantly smaller than a range of mean code lengths calculated via permutations, to test the mathematical theory behind both laws. In addition, we also fitted Bayesian generalized linear multivariate multilevel models with same number of iterations and chains as the previous models but having the median duration of each of the 26 gesture types as response variable, category of gesture as a fixed factor, as well as frequency of that gesture type as a predictor. These tests offer limited opportunities to control for potential confounds such as signaller identity and should be interpreted with caution in relatively small and variable datasets. We provide them in the electronic supplementary material, S6 to allow for comparison with previous work that analysed median durations with or without implementing generalized linear models (e.g. [4,36]).

Ethics. Ethical approval for this project was obtained from the University of St Andrews Animal Welfare and Ethics Committee under approval code PS15842 and all data collection was conducted under permission from the Ugandan Wildlife Authority and the Ugandan National Council for Science and Technology.

Data accessibility. Data and code for all analyses are available in a public GitHub repository: github.com/Wild-Minds/LinguisticLaws_Papers.

The data are provided in the electronic supplementary material [79].

Authors' contributions. A.S.: conceptualization, data curation, formal analysis, methodology, visualization, writing—original draft, writing—review and editing; C.C.: conceptualization, supervision, writing—review and editing; B.F.: data curation, investigation, writing—review and editing; R.H.: conceptualization, methodology, writing—review and editing; R.F.-i-C.: conceptualization, formal analysis, methodology, writing—original draft, writing—review and editing; C.H.: conceptualization, data curation, formal analysis, funding acquisition, investigation, methodology, project administration, supervision, writing—original draft, writing—review and editing.

All authors gave final approval for publication and agreed to be held accountable for the work performed therein.

Conflict of interest declaration. We declare we have no competing interests.

Funding. This research received funding from the European Union's 8th Framework Programme, Horizon 2020, under grant agreement no 802719.

Acknowledgements. We thank the staff and field assistants of the Budongo Conservation Field Station for their assistance in the original gestural data collection, and the Ugandan National Council for Science and Technology and the Ugandan Wildlife Authority for permission to conduct the original research. We thank the Royal Zoological Society of Scotland for its funding of the field station. We thank Dr Alexander Mielke for his advice on the statistical models. We thank the editor and three anonymous reviewers for their constructive comments.

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
