## [Peer Review File · Royal Society Open Science]

Review History

RSOS-211020.R0 (Original submission)

Review form: Reviewer 1

Is the manuscript scientifically sound in its present form?

No

Are the interpretations and conclusions justified by the results?

No

Is the language acceptable?

Yes

Do you have any ethical concerns with this paper?

No

Have you any concerns about statistical analyses in this paper?

No

Recommendation?

Reject

Comments to the Author(s)

This is a potentially interesting paper that is clearly written. However, I have serious concerns about the analyses that seem to limit the reliability of the conclusions.

My major concern is that it is not clear that units and sequences are defined the same way here as they are in vocal communication. While I appreciate the attempt to expand the application of linguistic patterns in new directions, I am not sure that what is presented here is really comparable. The main problem is that many of the gestures have quite long durations (more than 10 seconds) which does not seem at all comparable to units of speech (syllables or words). In looking at the gestures with long durations it appears that they are all ones that are likely to be repeated. For example, "object shake" can last for anywhere from <1 to 15 seconds. Surely that variation is not in the speed of a single shaking movement but rather is in the number of iterations of that movement. But repetitions of a movement are themselves a sequence. I understand that there might not be breaks in between (the movement is continuous) but I think the unit of analysis should be the smallest movement element that is repeated (a single movement in one direction or a cycle of back and forth?) and longer examples are considered multiple repetitions of that (so a sequence of X repeats each of short duration). Otherwise, I don't think it is really comparable to vocal communication. There are very few words that repeat the same syllable, and almost never more than once or twice. Bird song can be fairly continuous and last for several seconds but the unit of analysis for questions of compression would be notes or syllables within that song, not the entire song. Additionally, treating object shake (and the other repeated gestures) as single point in figure 1 is a little misleading. There were not >250 instances of a gesture that lasted about 2 seconds--there were a few instances of object shake across a wide range of durations (a range that is greater than the range across the means of all call types). The same problem applies to all of the analyses using average durations for gesture types--they are lumping together very different versions of the gestures, some with many repetitions, some with few. This is particularly problematic given that the repetitions themselves might be subject to the laws: e.g., are the shorter versions of object shake more common (Zipf's law, object shake was seemingly more common at shorter durations in figure S5) or are they more likely to appear in longer sequences with other gestures (Manzereth's law)? I know similar linguistic analyses use measures of average duration but the amount of variation within a type is much lower and not due to a variable number of repeats.

Another problem relating to the length of some of the gestures is the definition of the end point of the gesture. The second criteria is a change in position if the gesture involves a particular body alignment. This highlights another way that gestures are different from vocal production. Vocal production has to be actively produced--if you stop forcing air over the vocal cords, the sound stops. But a gesture that relies on a certain body alignment is entirely passive (once the posture is achieved). So it is unlikely that such gestures would be subject to the same mechanical pressures/costs. The time an animal is actively moving to produce the gesture is probably more directly comparable to vocal communication. Similarly, the last criteria seems to have more to do with the response of other's than the production of a signal. Response latency seems unrelated to compression and the mechanisms of signal production (response is more relevant to the function of the signal). Given these definitions, it does not seem that duration is entirely comparable across gestures and vocalizations, even for gestures that do not repeat.

It seems that the Menzerath finding (Figure 2, the compression test, the model in Table 3A) is not given much weight in the manuscript. Figure 2 looks like a pretty clear pattern (up to sequence size of 4 after which there are only 9 data points) but it is largely influenced by a single individual who has the most gestures in the data set (Duane). Taking Duane out of the data set removes the

pattern--but it also, by definition, greatly reduces the sample size (by about 1/2 it appears). Is this just a matter of a smaller sample failing to reach significance despite similar effect size? Is there a reason to remove Duane other than the fact that he had a lot of gestures? I guess I don't see any reason to think that removing Duane makes the data set better and not worse. Duane is described as an outlier but having the most data is not what it means to be an outlier. Also the significant model already contained ID as a random effect (Table 3A). The data set without Duane is given most of the weight in the abstract and discussion but I don't see a good justification for why. I understand removing him and showing that he has a strong influence on the result but I think that the finding based on the entire data set should get priority (in the absence of another reason to exclude Duane).

There appear to be a few inconsistencies in the figures. There are 5 whole body gestures in figure S5 but 6 in figure 1 (and the supplementary Table). In figure 1 it looks like 3 gestures occurred only 1 time but in S5 it looks like 4 did. In S5 some gestures (e.g., Beckon) appear to have a single point that does not match the median line.

Figure S5 is very useful and I appreciate the inclusion of the raw data. One suggestion for improving these kinds of figures is to arrange the X-axis in some meaningful way (rather than alphabetically) so patterns can be seen more quickly. For example, they could be arranged from highest to lowest median duration.

Review form: Reviewer 2

Is the manuscript scientifically sound in its present form?

Yes

Are the interpretations and conclusions justified by the results?

No

Is the language acceptable?

Yes

Do you have any ethical concerns with this paper?

No

Have you any concerns about statistical analyses in this paper?

Yes

Recommendation?

Major revision is needed (please make suggestions in comments)

Comments to the Author(s)

This is an interesting study that explores the possible existence of two linguistic laws – Zipf's law of brevity and Menzerath's law – in chimpanzee sexual solicitation gestural displays. It is great that the authors use this paper to build upon an earlier study on play gestures in the same study population. Building up a resource of longitudinal data like this is incredibly useful to the field of animal communication, and specifically, to the growing interest in applying linguistic laws to non-human behavior. My concerns are mostly related to the strong claims made in the manuscript despite there being major limitations in the dataset. This study is really more of a 'case study' that is focused on a small number of gestures from a single animal. I am still very enthusiastic about this study, and I'm sure that collecting these data was no easy task. However, I

do want to see some significant changes made to the paper so that it does a better job at representing the dataset and makes more conservative interpretations.

1. Generally speaking, the emphasis on null results comes across too strong given the study limitations. The results are based on a dataset that is 25% as big as the play gesture study that is referenced throughout the paper, there are only 12 gesture sequences over length 3, and the variation in sequence length is small ($n=6$ gestures, while the play study had up to ~ 16). The null results regarding Zipf's law of Brevity are somewhat convincing, but I suspect that null results involving menzerath law are an outcome of type II error. Figure 2 and the compression test analysis suggest that the law does apply. Also, over 50% of the dataset is from a single individual, which means that this study is more a case study than one that reflects population-level patterns. A dataset that is highly skewed towards one individual isn't very useful for assessing individual variation.

I have three suggestions here. (a) First, the limitations of the dataset should be made more transparent in the Methods/Results and Discussion, including how the dataset differs from the one on play gestures. (b) Second, throughout the manuscript, there should be a toning down of claims that the study showed that the laws "failed" (especially Menzerath law). There are many places where it is said that the laws "failed," and this phrasing is contentious. (c) Third, I recommend that the authors lean in to the "case study" aspect of their dataset rather than focusing on individual variation. In Figures 1 and 2, it would be helpful to include subplots that show raw data from the chimp with the majority of datapoints (Duane). In the main text or supplementary results, I'd like to see correlation/GLMM analyses only for Duane. In the Results and Discussion, there should be less emphasis on interpretations related to signaller effects, since the data are not distributed evenly across individuals.

2. Related to comment 1 above, it would help to show raw data so that readers can critique the statistical models and conclusions. I found it difficult to assess the model results because I did not understand the data structure. Given that the dataset is small, it should be easy to show it in figures. At a minimum, please show raw data from the animal with the most data in the figures on Zipf's law of Brevity and Menzerath's law. It would also help to add a descriptive figure that comes before the Zipf law figure. This figure could show what these gestures are and how they are combined together (e.g., cartoons of the gestures, repertoires of solo vs. sequence gestures, comparison of duration across gesture types and gesture positions with a sequence (1st, 2nd 3rd, etc)). If the authors need space, then move some tables to the supplement. For example, I found it confusing to have tables for BOTH AIC and BIC. Emphasize the most relevant method and move the other to the supplement?

3. I would like more rationale for why the authors chose the model formulas that they did. First, I do not understand the rationale for why gestures are split by "manual" vs "whole-body" (category in Model 1, PWB in Model 3) other than merely copying the methods from an earlier paper. Based on the supplemental information, it doesn't look like gestures in these categories differ much (i.e., one category isn't longer in duration or more stereotyped than the other). For Model 1, why not use "gesture type" (all 26 types) instead of a 2-level "category"? For Model 3, why not use gesture duration as the dependent variable (instead of average duration), and then include gesture type, sequence position, and sequence id as predictor/random effects? For both models 1 and 3, it seems odd that modified data measures (e.g., average duration, proportions) are used in GLMM analyses instead of the raw data variables. Why not use gesture frequency instead of proportions in Model 1? Why not use gesture token duration instead of averages? Why use PWB in model 3?

4. The introduction and discussion need more integration with recent perspectives on linguistic laws and what they mean. I recommend that the authors connect their study to discussions in a in

press review in TREE by Semple, Ferrer-i-Cancho and Gustison. This review synthesizes human and non-human literature, discusses what “universality” means, and links these laws to energy expenditure. There are also interesting discussions of these laws and evolutionary processes in papers by Torre and colleagues.

5. Related to the point above, I recommend that the authors more thoroughly discuss and reference recent human-focused work, since humans are a very relevant comparison to chimps. For example, any discussions of sign language and linguistic laws can be expanded (e.g., L51-110). Potential questions to answer include: Are both Zipf’s law of brevity AND menzerath’s law tested in human sign language studies? What were the conditions when these laws were tested in humans and are they at all analogous to the present study? (E.g., was sign language studied during naturalistic social interactions?). What different evolutionary pressures may be at play for human sign language vs. sexual solicitation gestures in chimps?

Minor comments

- Suggest rephrasing of the paper’s title. “Linguistic laws are not the law” is awkward. Only two linguistic laws are tested in this paper, but there are more than two linguistic laws. Also, there are only convincing results to argue against one of these – the Zipf’s law of brevity.

- L34: Suggest rephrasing of “pressure for efficiency that has been previously proposed to be universal”. This is “universality” argument is more nuanced than is suggested in the paper. This point is related to comments above that the authors should integrate more recent perspectives on what “universality” means (i.e., Semple et al. 2021).

- L36: Suggest rephrasing of “highlight that signallers consider signalling efficiency broadly”. This phrasing suggests that the chimpanzees are being intentional in how they choose gestures. This conclusion is not very convincing given this study does not get at intentionality. While I agree with the authors that intentionality is an intriguing possibility, it seems like other biological pressures related to arousal states or developmental constraints are more likely.

- L36-37: Unclear what is meant by “diverse factors”. Suggest being more specific here.

- L44-45: Rephrase “argued to be present across systems of biological information”. This argument is more nuanced than is presented in this manuscript. Again, I recommend the authors refer to the new TREE review by Semple and colleagues.

- L47: Rephrase “is an advantage to choosing the outcome”. This sounds intentional, and the study

- L81 (and throughout the manuscript): General use of the word “failure” comes across as contentious. I recommend that the authors tone this down. Typically, the authors are referring to instances when a study was unable to refute the null hypothesis (no correlation); a null result is less convincing than instances when a correlation is found in the opposing direction to a hypothesis.

- L88: Possible typo here, is Zipf law of “abbreviation” meant to be “brevity”?

- L103: Could use to develop this rationale a bit more. Why does it matter if sexual solicitations are urgent? Are they actually that urgent? What about energetic costs, what about these solicitations make them costly for males? Why does urgency matter?

- L109: Be more specific here about strong selection pressures. What kind of selection pressures? Is there evidence of female choice for more elaborate signals?
- L111-19: Could use more background information here about what is known about the composition of sexual solicitation gestures. How long are they? Do they vary within and across individuals? How do females respond to them (e.g., what parts of these gestures might be sensitive to selection pressures)?
- L119, L302, and L328: Suggest rephrasing of “novel evolutionarily urgent context” and “evolutionary salient scenarios”. It is not clear what these phrases actually mean. Does this mean that evolution happens quickly in sexual contexts?
- L125: Use median instead of mean since there is high skew in the dataset.
- L127-132: How many sequences per male?
- L141: Could use a sentence here describing what a “compression test” is. Not immediately obvious what this means.
- L142-144 (and throughout the manuscript): It was not clear to me at all what was meant in the terminology related to the compression tests. How is a gesture “big” or “small”? What does this mean? It is explained a bit in the Methods section, but could use a more general interpretation in the results section.
- L176-178: If BIC is more parsimonious, then why not just stick with that approach instead of including both? Personally, I find the inclusion of both AIC and BIC more confusing than helpful.
- L201-202: Why are the correlation test p-values different for the manual gestures across these two sentences? There should be one p-value for the positive correlation? If the first analysis is a one-tailed test and the second analysis is a two-tailed test, then please make this clear.
- L208 (and throughout): Rather than say “there was no relationship”, say “we found no evidence for a relationship”. Technically, correlation/GLMM tests are used to refute a null hypothesis, but they are not appropriate to “prove” that a null hypothesis exists. Also, on line 209, the correlation test ($p=0.076$) suggests that there is a trend in the hypothesized direction. Why isn’t this trend acknowledged? Marginal effects are acknowledged elsewhere in the paper (L205).
- Tables: The table titles could use more descriptive titles than “model 1” and “model 3”. How about including whether the models are testing Zipf’s law of brevity or Menzerath’s law?
- Where is model 2? The results and Tables appear to jump from 1 to 3. I assume that I missed some fine print stating that model 2 is in the supplement? This is confusing.
- L216-217: I did not understand this phrase “a linear association between n and t that could not be sufficiently captured by the Spearman correlation test.” Could use more explanation here. I thought that Spearman tests don’t assume that data are linear?
- L261: Suggest using more conservative phrasing instead of “... did not follow either Menzerath’s law or Zipf’s law of brevity”. This is a strong conclusion given that the dataset is limited and that correlation tests are not designed to prove a null hypothesis. It is strange to me that the authors say that there is no evidence for Menzerath’s law when Figure 2 looks quite convincing and the Spearman correlation shows a trend in the hypothesized direction. I suspect

that this was an instance of Type II error and not having enough data across sequence sizes and individuals.

- L263: Unclear what “no subsets” means. Explain.

- L270-279: This paragraph would also be a good place to develop reference to human sign language. One of these human studies is cited but is not discussed in any detail in the manuscript.

- L270-279 and L310-328: These are another places where links to Semple et al would be useful. “Universality” should be used in a more nuanced way than is depicted in this paragraph.

- L306+: The authors should consider adding in an experimental human study (Kanwal...Kirby 2017 Cognition) to this part of the discussion centered on “urgency” and in situ contextual variation. This human study tests the presence of zipf law of brevity when people are faced with pressures to produce efficient vs. accurate communication signals. Both pressures were needed for individuals to communicate in ways that supported Zipf’s law of abbreviation.

- L331-346: There needs to be more detail about when and how sexual solicitation gestures were recorded. What efforts were made to insure that males were observed as evenly as possible (e.g., focal observations, similar times of the day, etc)? How many females were involved in these recorded interactions, and were the same females involved across all males? How many gesture sequences were there and what was the range in sample sizes across males? What recording equipment was used? Were males always being recorded on video during focal samples? If video samples were only taken during sexual solicitations, then how often were gestures not caught on film because they occurred before the filming started?

- L359: This “1s” threshold to separate sequences is somewhat arbitrary. Can the authors provide a histogram of inter-gesture interval lengths to show that this threshold is appropriate for their study system?

- L429-441: The compression test approach could use more explanation. I still cannot understand it, especially what it means to be “big” or “small” (is this the same this as short or long duration?). It might help to show the observed parameters vs. permutated data as a panel in the relevant Results figures.

Review form: Reviewer 3

Is the manuscript scientifically sound in its present form?

Yes

Are the interpretations and conclusions justified by the results?

Yes

Is the language acceptable?

Yes

Do you have any ethical concerns with this paper?

No

Have you any concerns about statistical analyses in this paper?

Yes

Recommendation?

Major revision is needed (please make suggestions in comments)

Comments to the Author(s)

The paper is well-written and I enjoyed reading it. The data provides useful insights into chimpanzee gestural communication and the growing literature on the applicability of linguistic laws to animal communication. I think it will make a valuable addition to this literature.

However, there are a handful of issues which should be addressed before publication. I am confident these can mostly be addressed by providing additional details and argumentation, or streamlining the content. I have packaged these comments into the attached.pdf file (see Appendix A) for formatting reasons.

Decision letter (RSOS-211020.R0)

Dear Dr HOBATER

The Editors assigned to your paper RSOS-211020 "Linguistic laws are not the law in chimpanzee sexual solicitation gestures." have made a decision based on their reading of the paper and any comments received from reviewers.

Regrettably, in view of the reports received, the manuscript has been rejected in its current form. However, a new manuscript may be submitted which takes into consideration these comments.

We invite you to respond to the comments supplied below and prepare a resubmission of your manuscript. Below the referees' and Editors' comments (where applicable) we provide additional requirements. We provide guidance below to help you prepare your revision.

Please note that resubmitting your manuscript does not guarantee eventual acceptance, and we do not generally allow multiple rounds of revision and resubmission, so we urge you to make every effort to fully address all of the comments at this stage. If deemed necessary by the Editors, your manuscript will be sent back to one or more of the original reviewers for assessment. If the original reviewers are not available, we may invite new reviewers.

Please resubmit your revised manuscript and required files (see below) no later than 09-Jun-2022. Note: the ScholarOne system will 'lock' if resubmission is attempted on or after this deadline. If you do not think you will be able to meet this deadline, please contact the editorial office immediately.

Please note article processing charges apply to papers accepted for publication in Royal Society Open Science (<https://royalsocietypublishing.org/rsos/charges>). Charges will also apply to papers transferred to the journal from other Royal Society Publishing journals, as well as papers submitted as part of our collaboration with the Royal Society of Chemistry (<https://royalsocietypublishing.org/rsos/chemistry>). Fee waivers are available but must be requested when you submit your manuscript (<https://royalsocietypublishing.org/rsos/waivers>).

Thank you for submitting your manuscript to Royal Society Open Science and we look forward to receiving your resubmission. If you have any questions at all, please do not hesitate to get in touch.

on behalf of Dr Oliver Schülke (Associate Editor) and Kevin Padian (Subject Editor)
 openscience@royalsociety.org

Associate Editor Comments to Author (Dr Oliver Schülke):

Associate Editor: 1

Comments to the Author:

Dear Dr. Hobaiter,

my sincere apologies for taking so awfully long with responding to your submission. We had troubles finding reviewers, have lost two who had agreed along the way, but now can provide three very helpful reviews of your manuscript. All reviewers agreed that substantial work is needed before a final recommendation can be made - and it is possible that such recommendation may be negative. Comments by reviewer 1 on the fundamental definition of units and sequences challenge the idea that the linguistic laws can be applied to the gestural communication investigated here and these concerns are echoed in comments by reviewer 3 on the urgency of gestures, the role of intentionality, and the resulting direction in the two laws studied. All reviewers have issues with the statistical approach and the interpretation of the data that culminate in reviewer 2's suggestion to better view this as a detailed case study with supplemental information. If you feel that you can resolve the definition issues and clearly show how gestures and vocalizations are conceptually and biologically similar enough to warrant application of linguistic laws and answer to the comments on statistic analyses, we will consider a resubmitted manuscript with more carefully phrased conclusions.

Reviewer comments to Author:

Reviewer: 1

Comments to the Author(s)

This is a potentially interesting paper that is clearly written. However, I have serious concerns about the analyses that seem to limit the reliability of the conclusions.

My major concern is that it is not clear that units and sequences are defined the same way here as they are in vocal communication. While I appreciate the attempt to expand the application of linguistic patterns in new directions, I am not sure that what is presented here is really comparable. The main problem is that many of the gestures have quite long durations (more than 10 seconds) which does not seem at all comparable to units of speech (syllables or words). In looking at the gestures with long durations it appears that they are all ones that are likely to be repeated. For example, "object shake" can last for anywhere from <1 to 15 seconds. Surely that variation is in not in the speed of a single shaking movement but rather is in the number of iterations of that movement. But repetitions of a movement are themselves a sequence. I understand that there might not be breaks in between (the movement in continuous) but I think the unit of analysis should be the smallest movement element that is repeated (a single movement in one direction or a cycle of back and forth?) and longer examples are considered multiple repetitions of that (so a sequence of X repeats each of short duration). Otherwise, I don't

think it is really comparable to vocal communication. There are very few words that repeat the same syllable, and almost never more than once or twice. Bird song can be fairly continuous and last for several seconds but the unit of analysis for questions of compression would be notes or syllables within that song, not the entire song. Additionally, treating object shake (and the other repeated gestures) as single point in figure 1 is a little misleading. There were not >250 instances of a gesture that lasted about 2 seconds--there were a few instances of object shake across a wide range of durations (a range that is greater than the range across the means of all call types). The same problem applies to all of the analyses using average durations for gesture types--they are lumping together very different versions of the gestures, some with many repetitions, some with few. This is particularly problematic given that the repetitions themselves might be subject to the laws: e.g., are the shorter versions of object shake more common (Zipf's law, object shake was seemingly more common at shorter durations in figure S5) or are they more likely to appear in longer sequences with other gestures (Manzereth's law)? I know similar linguistic analyses use measures of average duration but the amount of variation within a type is much lower and not due to a variable number of repeats.

Another problem relating to the length of some of the gestures is the definition of the end point of the gesture. The second criteria is a change in position if the gesture involves a particular body alignment. This highlights another way that gestures are different from vocal production. Vocal production has to be actively produced--if you stop forcing air over the vocal cords, the sound stops. But a gesture that relies on a certain body alignment is entirely passive (once the posture is achieved). So it is unlikely that such gestures would be subject to the same mechanical pressures/costs. The time an animal is actively moving to produce the gesture is probably more directly comparable to vocal communication. Similarly, the last criteria seems to have more to do with the response of other's than the production of a signal. Response latency seems unrelated to compression and the mechanisms of signal production (response is more relevant to the function of the signal). Given these definitions, it does not seem that duration is entirely comparable across gestures and vocalizations, even for gestures that do not repeat.

It seems that the Menzerath finding (Figure 2, the compression test, the model in Table 3A) is not given much weight in the manuscript. Figure 2 looks like a pretty clear pattern (up to sequence size of 4 after which there are only 9 data points) but it is largely influenced by a single individual who has the most gestures in the data set (Duane). Taking Duane out of the data set removes the pattern--but it also, by definition, greatly reduces the sample size (by about 1/2 it appears). Is this just a matter of a smaller sample failing to reach significance despite similar effect size? Is there a reason to remove Duane other than the fact that he had a lot of gestures? I guess I don't see any reason to think that removing Duane makes the data set better and not worse. Duane is described as an outlier but having the most data is not what it means to be an outlier. Also the significant model already contained ID as a random effect (Table 3A). The data set without Duane is given most of the weight in the abstract and discussion but I don't see a good justification for why. I understand removing him and showing that he has a strong influence on the result but I think that the finding based on the entire data set should get priority (in the absence of another reason to exclude Duane).

There appear to be a few inconsistencies in the figures. There are 5 whole body gestures in figure S5 but 6 in figure 1 (and the supplementary Table). In figure 1 it looks like 3 gestures occurred only 1 time but in S5 it looks like 4 did. In S5 some gestures (e.g., Beckon) appear to have a single point that does not match the median line.

Figure S5 is very useful and I appreciate the inclusion of the raw data. One suggestion for improving these kinds of figures is to arrange the X-axis in some meaningful way (rather than alphabetically) so patterns can be seen more quickly. For example, they could be arranged from highest to lowest median duration.

Reviewer: 2

Comments to the Author(s) (see also attachment 'comments.pdf'):

This is an interesting study that explores the possible existence of two linguistic laws – Zipf’s law of brevity and Menzerath’s law – in chimpanzee sexual solicitation gestural displays. It is great that the authors use this paper to build upon an earlier study on play gestures in the same study population. Building up a resource of longitudinal data like this is incredibly useful to the field of animal communication, and specifically, to the growing interest in applying linguistic laws to non-human behavior. My concerns are mostly related to the strong claims made in the manuscript despite there being major limitations in the dataset. This study is really more of a ‘case study’ that is focused on a small number of gestures from a single animal. I am still very enthusiastic about this study, and I’m sure that collecting these data was no easy task. However, I do want to see some significant changes made to the paper so that it does a better job at representing the dataset and makes more conservative interpretations.

1. Generally speaking, the emphasis on null results comes across too strong given the study limitations. The results are based on a dataset that is 25% as big as the play gesture study that is referenced throughout the paper, there are only 12 gesture sequences over length 3, and the variation in sequence length is small ($n=6$ gestures, while the play study had up to ~ 16). The null results regarding Zipf’s law of Brevity are somewhat convincing, but I suspect that null results involving menzerath law are an outcome of type II error. Figure 2 and the compression test analysis suggest that the law does apply. Also, over 50% of the dataset is from a single individual, which means that this study is more a case study than one that reflects population-level patterns. A dataset that is highly skewed towards one individual isn’t very useful for assessing individual variation.

I have three suggestions here. (a) First, the limitations of the dataset should be made more transparent in the Methods/Results and Discussion, including how the dataset differs from the one on play gestures. (b) Second, throughout the manuscript, there should be a toning down of claims that the study showed that the laws “failed” (especially Menzerath law). There are many places where it is said that the laws “failed,” and this phrasing is contentious. (c) Third, I recommend that the authors lean in to the “case study” aspect of their dataset rather than focusing on individual variation. In Figures 1 and 2, it would be helpful to include subplots that show raw data from the chimp with the majority of datapoints (Duane). In the main text or supplementary results, I’d like to see correlation/GLMM analyses only for Duane. In the Results and Discussion, there should be less emphasis on interpretations related to signaller effects, since the data are not distributed evenly across individuals.

2. Related to comment 1 above, it would help to show raw data so that readers can critique the statistical models and conclusions. I found it difficult to assess the model results because I did not understand the data structure. Given that the dataset is small, it should be easy to show it in figures. At a minimum, please show raw data from the animal with the most data in the figures on Zipf’s law of Brevity and Menzerath’s law. It would also help to add a descriptive figure that comes before the Zipf law figure. This figure could show what these gestures are and how they are combined together (e.g., cartoons of the gestures, repertoires of solo vs. sequence gestures, comparison of duration across gesture types and gesture positions with a sequence (1st, 2nd 3rd, etc)). If the authors need space, then move some tables to the supplement. For example, I found it confusing to have tables for BOTH AIC and BIC. Emphasize the most relevant method and move the other to the supplement?

3. I would like more rationale for why the authors chose the model formulas that they did. First, I do not understand the rationale for why gestures are split by “manual” vs “whole-body” (category in Model 1, PWB in Model 3) other than merely copying the methods from an earlier

paper. Based on the supplemental information, it doesn't look like gestures in these categories differ much (i.e., one category isn't longer in duration or more stereotyped than the other). For Model 1, why not use "gesture type" (all 26 types) instead of a 2-level "category"? For Model 3, why not use gesture duration as the dependent variable (instead of average duration), and then include gesture type, sequence position, and sequence id as predictor/random effects? For both models 1 and 3, it seems odd that modified data measures (e.g., average duration, proportions) are used in GLMM analyses instead of the raw data variables. Why not use gesture frequency instead of proportions in Model 1? Why not use gesture token duration instead of averages? Why use PWB in model 3?

4. The introduction and discussion need more integration with recent perspectives on linguistic laws and what they mean. I recommend that the authors connect their study to discussions in a in press review in TREE by Semple, Ferrer-i-Cancho and Gustison. This review synthesizes human and non-human literature, discusses what "universality" means, and links these laws to energy expenditure. There are also interesting discussions of these laws and evolutionary processes in papers by Torre and colleagues.

5. Related to the point above, I recommend that the authors more thoroughly discuss and reference recent human-focused work, since humans are a very relevant comparison to chimps. For example, any discussions of sign language and linguistic laws can be expanded (e.g., L51-110). Potential questions to answer include: Are both Zipf's law of brevity AND menzerath's law tested in human sign language studies? What were the conditions when these laws were tested in humans and are they at all analogous to the present study? (E.g., was sign language studied during naturalistic social interactions?). What different evolutionary pressures may be at play for human sign language vs. sexual solicitation gestures in chimps?

Minor comments

- Suggest rephrasing of the paper's title. "Linguistic laws are not the law" is awkward. Only two linguistic laws are tested in this paper, but there are more than two linguistic laws. Also, there are only convincing results to argue against one of these – the Zipf's law of brevity.

- L34: Suggest rephrasing of "pressure for efficiency that has been previously proposed to be universal". This is "universality" argument is more nuanced than is suggested in the paper. This point is related to comments above that the authors should integrate more recent perspectives on what "universality" means (i.e., Semple et al. 2021).

- L36: Suggest rephrasing of "highlight that signallers consider signalling efficiency broadly". This phrasing suggests that the chimpanzees are being intentional in how they choose gestures. This conclusion is not very convincing given this study does not get at intentionality. While I agree with the authors that intentionality is an intriguing possibility, it seems like other biological pressures related to arousal states or developmental constraints are more likely.

- L36-37: Unclear what is meant by "diverse factors". Suggest being more specific here.

- L44-45: Rephrase "argued to be present across systems of biological information". This argument is more nuanced than is presented in this manuscript. Again, I recommend the authors refer to the new TREE review by Semple and colleagues.

- L47: Rephrase "is an advantage to choosing the outcome". This sounds intentional, and the study

- L81 (and throughout the manuscript): General use of the word "failure" comes across as contentious. I recommend that the authors tone this down. Typically, the authors are referring to

instances when a study was unable to refute the null hypothesis (no correlation); a null result is less convincing than instances when a correlation is found in the opposing direction to a hypothesis.

- L88: Possible typo here, is Zipf law of “abbreviation” meant to be “brevity”?

- L103: Could use to develop this rationale a bit more. Why does it matter if sexual solicitations are urgent? Are they actually that urgent? What about energetic costs, what about these solicitations make them costly for males? Why does urgency matter?

- L109: Be more specific here about strong selection pressures. What kind of selection pressures? Is there evidence of female choice for more elaborate signals?

- L111-19: Could use more background information here about what is known about the composition of sexual solicitation gestures. How long are they? Do they vary within and across individuals? How do females respond to them (e.g., what parts of these gestures might be sensitive to selection pressures)?

- L119, L302, and L328: Suggest rephrasing of “novel evolutionarily urgent context” and “evolutionary salient scenarios”. It is not clear what these phrases actually mean. Does this mean that evolution happens quickly in sexual contexts?

- L125: Use median instead of mean since there is high skew in the dataset.

- L127-132: How many sequences per male?

- L141: Could use a sentence here describing what a “compression test” is. Not immediately obvious what this means.

- L142-144 (and throughout the manuscript): It was not clear to me at all what was meant in the terminology related to the compression tests. How is a gesture “big” or “small”? What does this mean? It is explained a bit in the Methods section, but could use a more general interpretation in the results section.

- L176-178: If BIC is more parsimonious, then why not just stick with that approach instead of including both? Personally, I find the inclusion of both AIC and BIC more confusing than helpful.

- L201-202: Why are the correlation test p-values different for the manual gestures across these two sentences? There should be one p-value for the positive correlation? If the first analysis is a one-tailed test and the second analysis is a two-tailed test, then please make this clear.

- L208 (and throughout): Rather than say “there was no relationship”, say “we found no evidence for a relationship”. Technically, correlation/GLMM tests are used to refute a null hypothesis, but they are not appropriate to “prove” that a null hypothesis exists. Also, on line 209, the correlation test ($p=0.076$) suggests that there is a trend in the hypothesized direction. Why isn't this trend acknowledged? Marginal effects are acknowledged elsewhere in the paper (L205).

- Tables: The table titles could use more descriptive titles than “model 1” and “model 3”. How about including whether the models are testing Zipf's law of brevity or menzerath's law?

- Where is model 2? The results and Tables appear to jump from 1 to 3. I assume that I missed some fine print stating that model 2 is in the supplement? This is confusing.

- L216-217: I did not understand this phrase “a linear association between n and t that could not be sufficiently captured by the Spearman correlation test.” Could use more explanation here. I thought that spearman tests don't assume that data are linear?

- L261: Suggest using more conservative phrasing instead of “... did not follow either Menzerath's law or Zipf's law of brevity”. This is a strong conclusion given that the dataset is limited and that correlation tests are not designed to prove a null hypothesis. It is strange to me that the authors say that there is no evidence for Menzerath's law when Figure 2 looks quite convincing and the Spearman correlation shows a trend in the hypothesized direction. I suspect that this was an instance of Type II error and not having enough data across sequence sizes and individuals.

- L263: Unclear what “no subsets” means. Explain.

- L270-279: This paragraph would also be a good place to develop reference to human sign language. One of these human studies is cited but is not discussed in any detail in the manuscript.

- L270-279 and L310-328: These are another places where links to Semple et al would be useful. “Universality” should be used in a more nuanced way than is depicted in this paragraph.

- L306+: The authors should consider adding in an experimental human study (Kanwal...Kirby 2017 Cognition) to this part of the discussion centered on “urgency” and in situ contextual variation. This human study tests the presence of zipf law of brevity when people are faced with pressures to produce efficient vs. accurate communication signals. Both pressures were needed for individuals to communicate in ways that supported Zipf's law of abbreviation.

- L331-346: There needs to be more detail about when and how sexual solicitation gestures were recorded. What efforts were made to insure that males were observed as evenly as possible (e.g., focal observations, similar times of the day, etc)? How many females were involved in these recorded interactions, and were the same females involved across all males? How many gesture sequences were there and what was the range in sample sizes across males? What recording equipment was used? Were males always being recorded on video during focal samples? If video samples were only taken during sexual solicitations, then how often were gestures not caught on film because they occurred before the filming started?

- L359: This “1s” threshold to separate sequences is somewhat arbitrary. Can the authors provide a histogram of inter-gesture interval lengths to show that this threshold is appropriate for their study system?

- L429-441: The compression test approach could use more explanation. I still cannot understand it, especially what it means to be “big” or “small” (is this the same this as short or long duration?). It might help to show the observed parameters vs. permutated data as a panel in the relevant Results figures.

Reviewer: 3

Comments to the Author(s) (see also 'RSOS Review 061221.pdf' attached):

The paper is well-written and I enjoyed reading it. The data provides useful insights into chimpanzee gestural communication and the growing literature on the applicability of linguistic laws to animal communication. I think it will make a valuable addition to this literature.

However, there are a handful of issues which should be addressed before publication. I am confident these can mostly be addressed by providing additional details and argumentation, or

streamlining the content. I have packaged these comments into the attached .pdf file for formatting reasons.

===PREPARING YOUR MANUSCRIPT===

If you have been asked to revise the written English in your submission as a condition of publication, you must do so, and you are expected to provide evidence that you have received language editing support. The journal would prefer that you use a professional language editing service and provide a certificate of editing, but a signed letter from a colleague who is a fluent speaker of English is acceptable. Note the journal has arranged a number of discounts for authors using professional language editing services (<https://royalsociety.org/journals/authors/benefits/language-editing/>).

===PREPARING YOUR REVISION IN SCHOLARONE===

<https://royalsociety.org/journals/authors/author-guidelines/#supplementary-material> to include a suitable title and informative caption. An example of appropriate titling and captioning may be found at [https://figshare.com/articles/Table_S2_from_Is_there_a_trade-off_between_peak_performance_and_performance_breadth_across_temperatures_for_aerobic_sc ope_in_teleost_fishes_/3843624](https://figshare.com/articles/Table_S2_from_Is_there_a_trade-off_between_peak_performance_and_performance_breadth_across_temperatures_for_aerobic_scope_in_teleost_fishes_/3843624).

Author's Response to Decision Letter for (RSOS-211020.R0)

See Appendix B.

RSOS-220849.R0

Review form: Reviewer 1

Is the manuscript scientifically sound in its present form?

Yes

Are the interpretations and conclusions justified by the results?

Yes

Is the language acceptable?

Yes

Do you have any ethical concerns with this paper?

No

Have you any concerns about statistical analyses in this paper?

No

Recommendation?

Accept with minor revision (please list in comments)

Comments to the Author(s)

I appreciate the extensive revisions and I think the manuscript is improved. I have two lingering concerns. First, while the focus on the gestures from Duane is improved, I am not sure it is handled in the best way. To me, the question is, do the gestures of Duane show a different pattern than the gestures from everyone else? To answer that, you would want to run the analyses on the other (non-Duane) gestures alone. Analyzing all the gestures together and separating out Duane does not answer this (all gestures together may show the pattern because it is being driven by the large number of Duane samples – the reader can not tell). I can piece the story together across the revisions (because, if I recall correctly, in the previous version the non-Duane samples did not show the pattern) but it should be explored here (with the new analysis methods). What would it mean to have the pattern show up in one sample but not the other? I think it is ok to have the story be somewhat complicated. My second issue is one I raised previously – that repeated (and held postural gestures) may be different from other units analyzed in sequences. Previously I had pointed out that they are not like vocal sequences – and I agree with the response that it makes sense to look at these laws in all kinds of systems, not just vocal sequences. However, repeated (and held gestures) are not like the other gestures that they are being compared to within this manuscript (some of which seem to be constrained to a short duration). I did not make this point clear in my previous review but this could create problems beyond the theoretical issue of mixing sequences (in the case of repeated gestures) and units in the same analysis. For example, it could be that the ‘prone to long duration’ gestures (repeated or held) are rarely embedded in longer sequences. This would produce the Menzereth pattern but not because of ‘compression’, rather because of different uses of different gesture types. This seems relatively easy to test for – is there any evidence of Menzereth occurring within the common gesture types? Does separating the ‘prone to long duration’ call types change things?

Review form: Reviewer 2

Is the manuscript scientifically sound in its present form?

No

Are the interpretations and conclusions justified by the results?

No

Is the language acceptable?

Yes

Do you have any ethical concerns with this paper?

No

Have you any concerns about statistical analyses in this paper?

Yes

Recommendation?

Major revision is needed (please make suggestions in comments)

Comments to the Author(s)

The authors did a great job at addressing my feedback. This version of the manuscript is streamlined, and most of the results are communicated clearly. At this stage, I only have comments related to the Zipf law analysis and the Figures. I am confused by how Zipf's law was tested in this version of the manuscript. In addition, there is little visual representation of the data analyses, which makes it hard to evaluate statistical output and interpretations. These comments should be straightforward to address.

1. Zipf Law analysis: Zipf's law of brevity is simply defined as a negative association between unit frequency and unit duration. In this dataset, units are gesture types ($n=26$), and the (mean or median) duration and frequency of occurrence for each of these 26 gestures should be plotted against each other. It doesn't seem like the revised analysis approach directly tests Zipf's law of brevity, and it's unclear to me what is actually being tested. Based on the S4 and S5 tables, it looks like Proportions (for all gestures per all sequences?) were used as a predictor variable, and Gesture ID was a random effect variable. Therefore, there are 560 (or 290) datapoints, instead of 26. To directly test Zipf's law of brevity, I recommend using a Bayesian model with 26 data points, where gesture type duration is predicted by gesture type frequency, with gesture category (whole body, manual) included as another predictor variable or as a random effect.
2. Figure 1: There doesn't seem to be a visual representation of the analysis related to Zipf law. I recommend that the authors add a subplot or two to Figure 1 to show what the data for these analyses (Table S4 and S5) looked like. I suspect this plot would look something like Figure 2-3, where there is a full dataset and a Duane-only subset (see comments for Figure 2-3). Something similar to Figure 1 from the initial submission of the manuscript would be great.
3. Figure 2-3: Figure 3 is essentially a repetition of Figure 2. I recommend that the authors combine these figures since they are the same dataset and analysis approach.
4. In all Figures, please include results of statistical tests where appropriate inside the figures themselves. When using a Bayesian model approach, the regression line (slope and intercept from the model) can be overlaid on top of the data. In addition, the significance level of the test can be included on the figure.
5. In all Figures, please define the axes and units of measurement in the legends. For example, in Figures 2-3, it is not clear what "sequence size" means and what the points represent (each point = gesture? Or each point = sequence?).
6. Additional supplemental figure: It would be helpful to provide a couple figures showing individual animal data related to Zipf-model and Menzerath-model (e.g., 16-panel figures similar to Figure S8 in initial submission). Providing such figures would allow readers to understand the individual variation in the data, which has become an important issue in some recent papers.

Review form: Reviewer 3

Is the manuscript scientifically sound in its present form?

Yes

Are the interpretations and conclusions justified by the results?

Yes

Is the language acceptable?

Yes

Do you have any ethical concerns with this paper?

No

Have you any concerns about statistical analyses in this paper?

Yes

Recommendation?

Accept with minor revision (please list in comments)

Comments to the Author(s)

I commend the authors for the hard work they have put into their substantial revisions on this manuscript. They have dealt with my major conceptual issues with the original manuscript, and the statistical analysis is now much easier to follow. I have just two concerns about this revised analysis which I describe below:

1) The authors now acknowledge that the great majority of their data comes from a single individual ('Duane') and have changed their analysis as a result: Analysing all of the individuals together, and Duane separately. The authors claim that doing so validates the generalisability of their findings, but I am not sure what this means.

"As our data may be particularly influenced by a single prolific individual (Duane) who contributed around half of the data, we assess the generalizability of our findings by replicating analyses conducted on the full dataset on a subset of the data containing only gestures by Duane."

How exactly pooling Duane's data with the other individuals tells us about the generalisability of these findings is not clear at all in terms of statistical inference. The authors need to unpack their argumentation much more on this matter.

"As a result we consider it a case-study; nevertheless, our findings were similar for both the full dataset across male signallers, and for a single prolific individual, as well as in a range of alternative analyses (Supporting Information 1), suggesting that the pattern of results appears to be relatively robust."

I am not convinced. When there is an effect present in only Duane's data, the fact that a weaker effect is also found in a dataset where he contributes over half of the datapoints, is unsurprising. To convincingly show that the effect is not entirely driven by Duane, the authors would surely need to demonstrate that the effect is present in a dataset that does -not- include Duane's data?

2) I am also concerned that Duane's data was not analysed appropriately when examined individually.

Line 452: “For the models testing Duane’s data, signaller ID was removed from the random factors.”

A GLMM using 260 datapoints from a single individual, without random effects, will treat this as 1 datapoint from each of 260 individuals (repeating the pseudoreplication issue I highlighted in my previous review), falsely increasing the ‘power’ of the model. Including a random effect with one level is also rather meaningless, so I think an entirely different approach is needed to analyse Duane’s data effectively. Unfortunately, I am not practiced enough in analysing $N = 1$ data to recommend an alternative, so expert statistical advice should be sought.

Having said all this, the fact that the relationship holds when Duane’s data is pooled with the others and a random effect of ID is introduced does alleviate my concerns somewhat.

Minor comments

Line 33: I suggest replacing “fail to find” with “did not find”, as the former implies not being able to find something that is actually there, or that we at least wanted to be there.

Lines 116-117: These sentences do not really make sense to me. i) one explanation for what? ii) what case of long-distance calls? iii) “impacted its expression” is very vague, iv) “in this case” which case?

Line 136 & 278: As in my previous review, I urge the authors to nail down the meaning of the term “urgent” or else not use it at all.

Line 156: Unclear how a solicitation gesture might lead to lethal aggression from a neighbouring group. Do the authors mean when attempting solicitation with a female *from* a neighbouring group?

Line 164: I find the addition of this methods primer to the Introduction very useful for this format of manuscript. However, the movement between tenses (“we test” \ “we will test” \ “we tested”) through the manuscript is not ideal. The tests have already been carried out - it is only their presentation to the audience that has not occurred yet.

Line 210: What was the R^2 value? It would be just as important that it is not < 1 . Just give the raw value, it should be $= 1$ in all cases (or else there is an issue with convergence).

Line 223: “ R^2 values < 1.02 .”

Same issue as above.

Line 223: the term “significant effect” is not really appropriate when applying Bayesian methods. Please rephrase throughout. More importantly, the effect reported does not seem to be ‘significant’ as it has been reported:

“ $b = -0.18$, s.d. = 0.04 , 95% CrI [0.26 , -0.11]; Figure 2).”

For a robust effect I would expect to see credible intervals that do not overlap at all with 0. I see from inspecting the supplementary material that ‘0.26’ is simply missing a minus symbol. Please insert.

Indeed, there are a number of typos throughout the manuscript that have crept in with the revised text. I encourage the authors to carry out a careful proof-read.

Line 336: "while many vocalisations are relatively fixed"
Fixed in what sense? Support with references.

Line 423: Apologies for not picking up on this in the last draft, but one would ideally expect to see inter-observer reliability testing reported. While it's important that raters are consistent with themselves, a consistently incorrect rater is also of no use.

Line 446-7: "Random factors" - random intercepts, random slopes, or both? Given the focus of the analysis on determining whether Duane is an outlier or representative of the general population, I imagine random slopes would be appropriate to allow for individual variation.

Decision letter (RSOS-220849.R0)

Dear Dr Hobaiter,

The Editors assigned to your paper RSOS-220849 "Variable expression of linguistic laws in ape gesture: a case study from chimpanzee sexual solicitation." have now received comments from reviewers and would like you to revise the paper in accordance with the reviewer comments and any comments from the Editors. Please note this decision does not guarantee eventual acceptance.

Please submit your revised manuscript and required files (see below) no later than 21 days from today's (ie 11-Aug-2022) date. Note: the ScholarOne system will 'lock' if submission of the revision is attempted 21 or more days after the deadline. If you do not think you will be able to meet this deadline please contact the editorial office immediately.

on behalf of Dr Oliver Schülke (Associate Editor) and Kevin Padian (Subject Editor)
openscience@royalsociety.org

Associate Editor Comments to Author (Dr Oliver Schülke):

Dear Dr. Hobaiter,

Your original reviewers were kind enough to provide a second round of comments. In view of these comments I suggest major revisions to address the issues raised including additional analyses and plots. I am confident these revisions will significantly add to the accessibility of your research and look forward to receiving a new version of your contribution.

Best wishes,
Oliver Schülke

Reviewer comments to Author:

Reviewer: 1

Comments to the Author(s)

I appreciate the extensive revisions and I think the manuscript is improved. I have two lingering concerns. First, while the focus on the gestures from Duane is improved, I am not sure it is handled in the best way. To me, the question is, do the gestures of Duane show a different pattern than the gestures from everyone else? To answer that, you would want to run the analyses on the other (non-Duane) gestures alone. Analyzing all the gestures together and separating out Duane does not answer this (all gestures together may show the pattern because it is being driven by the large number of Duane samples – the reader can not tell). I can piece the story together across the revisions (because, if I recall correctly, in the previous version the non-Duane samples did not show the pattern) but it should be explored here (with the new analysis methods). What would it mean to have the pattern show up in one sample but not the other? I think it is ok to have the story be somewhat complicated. My second issue is one I raised previously – that repeated (and held postural gestures) may be different from other units analyzed in sequences. Previously I had pointed out that they are not like vocal sequences – and I agree with the response that it makes sense to look at these laws in all kinds of systems, not just vocal sequences. However, repeated (and held gestures) are not like the other gestures that they are being compared to within this manuscript (some of which seem to be constrained to a short duration). I did not make this point clear in my previous review but this could create problems beyond the theoretical issue of mixing sequences (in the case of repeated gestures) and units in the same analysis. For example, it could be that the ‘prone to long duration’ gestures (repeated or held) are rarely embedded in longer sequences. This would produce the Menzereth pattern but not because of ‘compression’, rather because of different uses of different gesture types. This seems relatively easy to test for – is there any evidence of Menzereth occurring within the common gesture types? Does separating the ‘prone to long duration’ call types change things?

Reviewer: 2

Comments to the Author(s)

The authors did a great job at addressing my feedback. This version of the manuscript is streamlined, and most of the results are communicated clearly. At this stage, I only have comments related to the Zipf law analysis and the Figures. I am confused by how Zipf’s law was tested in this version of the manuscript. In addition, there is little visual representation of the data analyses, which makes it hard to evaluate statistical output and interpretations. These comments should be straightforward to address.

1. Zipf Law analysis: Zipf's law of brevity is simply defined as a negative association between unit frequency and unit duration. In this dataset, units are gesture types ($n=26$), and the (mean or median) duration and frequency of occurrence for each of these 26 gestures should be plotted against each other. It doesn't seem like the revised analysis approach directly tests Zipf's law of brevity, and it's unclear to me what is actually being tested. Based on the S4 and S5 tables, it looks like Proportions (for all gestures per all sequences?) were used as a predictor variable, and Gesture ID was a random effect variable. Therefore, there are 560 (or 290) datapoints, instead of 26. To directly test Zipf's law of brevity, I recommend using a Bayesian model with 26 data points, where gesture type duration is predicted by gesture type frequency, with gesture category (whole body, manual) included as another predictor variable or as a random effect.

2. Figure 1: There doesn't seem to be a visual representation of the analysis related to Zipf law. I recommend that the authors add a subplot or two to Figure 1 to show what the data for these analyses (Table S4 and S5) looked like. I suspect this plot would look something like Figure 2-3, where there is a full dataset and a Duane-only subset (see comments for Figure 2-3). Something similar to Figure 1 from the initial submission of the manuscript would be great.

3. Figure 2-3: Figure 3 is essentially a repetition of Figure 2. I recommend that the authors combine these figures since they are the same dataset and analysis approach.

4. In all Figures, please include results of statistical tests where appropriate inside the figures themselves. When using a Bayesian model approach, the regression line (slope and intercept from the model) can be overlaid on top of the data. In addition, the significance level of the test can be included on the figure.

5. In all Figures, please define the axes and units of measurement in the legends. For example, in Figures 2-3, it is not clear what "sequence size" means and what the points represent (each point = gesture? Or each point = sequence?).

6. Additional supplemental figure: It would be helpful to provide a couple figures showing individual animal data related to Zipf-model and Menzerath-model (e.g., 16-panel figures similar to Figure S8 in initial submission). Providing such figures would allow readers to understand the individual variation in the data, which has become an important issue in some recent papers.

Reviewer: 3

Comments to the Author(s)

I commend the authors for the hard work they have put into their substantial revisions on this manuscript. They have dealt with my major conceptual issues with the original manuscript, and the statistical analysis is now much easier to follow. I have just two concerns about this revised analysis which I describe below:

1) The authors now acknowledge that the great majority of their data comes from a single individual ('Duane') and have changed their analysis as a result: Analysing all of the individuals together, and Duane separately. The authors claim that doing so validates the generalisability of their findings, but I am not sure what this means.

"As our data may be particularly influenced by a single prolific individual (Duane) who contributed around half of the data, we assess the generalizability of our findings by replicating analyses conducted on the full dataset on a subset of the data containing only gestures by Duane."

How exactly pooling Duane's data with the other individuals tells us about the generalisability of these findings is not clear at all in terms of statistical inference. The authors need to unpack their argumentation much more on this matter.

"As a result we consider it a case-study; nevertheless, our findings were similar for both the full dataset across male signallers, and for a single prolific individual, as well as in a range of alternative analyses (Supporting Information 1), suggesting that the pattern of results appears to be relatively robust."

I am not convinced. When there is an effect present in only Duane's data, the fact that a weaker effect is also found in a dataset where he contributes over half of the datapoints, is unsurprising. To convincingly show that the effect is not entirely driven by Duane, the authors would surely need to demonstrate that the effect is present in a dataset that does -not- include Duane's data?

2) I am also concerned that Duane's data was not analysed appropriately when examined individually.

Line 452: "For the models testing Duane's data, signaller ID was removed from the random factors."

A GLMM using 260 datapoints from a single individual, without random effects, will treat this as 1 datapoint from each of 260 individuals (repeating the pseudoreplication issue I highlighted in my previous review), falsely increasing the 'power' of the model. Including a random effect with one level is also rather meaningless, so I think an entirely different approach is needed to analyse Duane's data effectively. Unfortunately, I am not practiced enough in analysing $N = 1$ data to recommend an alternative, so expert statistical advice should be sought.

Having said all this, the fact that the relationship holds when Duane's data is pooled with the others and a random effect of ID is introduced does alleviate my concerns somewhat.

Minor comments

Line 33: I suggest replacing "fail to find" with "did not find", as the former implies not being able to find something that is actually there, or that we at least wanted to be there.

Lines 116-117: These sentences do not really make sense to me. i) one explanation for what? ii) what case of long-distance calls? iii) "impacted its expression" is very vague, iv) "in this case" which case?

Line 136 & 278: As in my previous review, I urge the authors to nail down the meaning of the term "urgent" or else not use it at all.

Line 156: Unclear how a solicitation gesture might lead to lethal aggression from a neighbouring group. Do the authors mean when attempting solicitation with a female *from* a neighbouring group?

Line 164: I find the addition of this methods primer to the Introduction very useful for this format of manuscript. However, the movement between tenses ("we test" \ "we will test" \ "we tested") through the manuscript is not ideal. The tests have already been carried out - it is only their presentation to the audience that has not occurred yet.

Line 210: What was the R^2 value? It would be just as important that it is not <1 . Just give the raw value, it should be = 1 in all cases (or else there is an issue with convergence).

Line 223: “ R^2 values < 1.02 .”

Same issue as above.

Line 223: the term “significant effect” is not really appropriate when applying Bayesian methods. Please rephrase throughout. More importantly, the effect reported does not seem to be ‘significant’ as it has been reported:

“ $b = -0.18$, $s.d. = 0.04$, 95% CrI [0.26, -0.11]; Figure 2).”

For a robust effect I would expect to see credible intervals that do not overlap at all with 0. I see from inspecting the supplementary material that ‘0.26’ is simply missing a minus symbol. Please insert.

Indeed, there are a number of typos throughout the manuscript that have crept in with the revised text. I encourage the authors to carry out a careful proof-read.

Line 336: “while many vocalisations are relatively fixed”

Fixed in what sense? Support with references.

Line 423: Apologies for not picking up on this in the last draft, but one would ideally expect to see inter-observer reliability testing reported. While it’s important that raters are consistent with themselves, a consistently incorrect rater is also of no use.

Line 446-7: “Random factors” – random intercepts, random slopes, or both? Given the focus of the analysis on determining whether Duane is an outlier or representative of the general population, I imagine random slopes would be appropriate to allow for individual variation.

===PREPARING YOUR MANUSCRIPT===

a ‘clean’ version of the new manuscript that incorporates the changes made, but does not highlight them. This version will be used for typesetting if your manuscript is accepted.

Please ensure that you include an acknowledgements’ section before your reference list/bibliography. This should acknowledge anyone who assisted with your work, but does not qualify as an author per the guidelines at <https://royalsociety.org/journals/ethics-policies/openness/>.

If you have been asked to revise the written English in your submission as a condition of publication, you must do so, and you are expected to provide evidence that you have received

language editing support. The journal would prefer that you use a professional language editing service and provide a certificate of editing, but a signed letter from a colleague who is a fluent speaker of English is acceptable. Note the journal has arranged a number of discounts for authors using professional language editing services (<https://royalsociety.org/journals/authors/benefits/language-editing/>).

===PREPARING YOUR REVISION IN SCHOLARONE===

<https://royalsociety.org/journals/authors/author-guidelines/#supplementary-material> to

include a suitable title and informative caption. An example of appropriate titling and captioning may be found at https://figshare.com/articles/Table_S2_from_Is_there_a_trade-off_between_peak_performance_and_performance_breadth_across_temperatures_for_aerobic_scope_in_teleost_fishes_/3843624.

Author's Response to Decision Letter for (RSOS-220849.R0)

See Appendix C.

Decision letter (RSOS-220849.R1)

Dear Dr HOBAITER:

I am pleased to inform you that your manuscript entitled "Variable expression of linguistic laws in ape gesture: a case study from chimpanzee sexual solicitation." is now accepted for publication in Royal Society Open Science.

Please remember to make any data sets or code libraries 'live' prior to publication, and update any links as needed when you receive a proof to check - for instance, from a private 'for review' URL to a publicly accessible 'for publication' URL. It is also good practice to add data sets, code and other digital materials to your reference list.

Royal Society Open Science is a fully open access journal. A payment may be due before your article is published. Our partner Copyright Clearance Center's RightsLink for Scientific Communications will contact the corresponding author about your open access options from the email domain @copyright.com (if you have any queries regarding fees, please see <https://royalsocietypublishing.org/rsos/charges> or contact authorfees@royalsociety.org).

Please see the Royal Society Publishing guidance on how you may share your accepted author manuscript at <https://royalsociety.org/journals/ethics-policies/media-embargo/>. After publication, some additional ways to effectively promote your article can also be found here

<https://royalsociety.org/blog/2020/07/promoting-your-latest-paper-and-tracking-your-results/>.

on behalf of Dr Oliver Schülke (Associate Editor) and Professor Kevin Padian (Subject Editor).

Associate Editor Dr Oliver Schülke Comments to Author:
Comments to the Author:

Many thanks to the authors for another round of thorough revisions and for the open discussion of the IOR issue. I feel that the manuscript has been improved a lot in both rounds of revisions and is ready for publication now.

Follow Royal Society Publishing on Twitter: @RSocPublishing
Follow Royal Society Publishing on Facebook:
<https://www.facebook.com/RoyalSocietyPublishing/>
Read Royal Society Publishing's blog:
<https://royalsociety.org/blog/blogsearchpage/?category=Publishing>

Appendix A

Firstly, there are a few conceptual points which I believe the manuscript would greatly benefit from having unpacked a bit further by the authors:

Conceptual issues

The paper often refers to the fact that sexual solicitation gestures are ‘urgent’, and I think this needs to be unpacked a bit more. Solicitation gestures are ‘urgent’ in the sense that unsuccessful solicitation means you are somewhat less likely to have offspring over the course of your lifetime, but this is very different to the kind of *temporal* urgency inherent to e.g. alarm signals (where things need to happen quickly or else you get eaten). It’s therefore not immediately clear why a gesture with a long duration would impact reproductive success (do chimps always wait until a gesture is finished before responding? Why is a fast response important?), whereas a prolonged long alarm call has obvious potential fitness consequences. Indeed, in birds, for example, more elaborate sexual displays are more successful.

The authors touch on this at points, hinting that fast gestures may be easier to conceal from dominant rivals. However, this is not factored into their analysis (e.g. fitting rank as a predictor, since dominant individuals should not experience pressure towards compression), and does not sit well with the fact that these gestures often seem to occur in sequences (making them more salient again). My understanding of the literature is that gestural sequences in chimps are primarily persistence and elaboration on the initial signal (used as evidence of intentionality) – my prediction would then be that gestures later in a sequence might have a longer duration than those that come earlier, as they become increasingly exaggerated to elicit the intended behavioural response.

e.g. **Line 303**: “the costs of signal failure are high, there is a pressure against compression and towards redundancy” – this makes absolute sense for signals such as alarm calls where the costs of not being heard are direct, and the benefits of redundancy are clear. However, it is not clear precisely how this applies to sexual solicitation gestures, and needs to be unpacked by the authors. E.g. what is failure constitutes in this context, and why redundancy would mitigate it. If failure is simply not having your signal recognised, then there is near-zero cost (except the very minor energetic cost of producing the signal) – you can simply try again. If failure is a prospective mate seeing your signal and turning down copulation, it is unclear how redundancy would mitigate this.

Comments on the Statistical analysis

While my comments on this section are quite extensive, I wish to emphasise that I do not think the analysis has necessarily been carried out incorrectly. Indeed, I broadly agree with the authors’ interpretation of the data. My comments and suggestions largely concern eliminating redundancy and improving clarity to improve the reader experience. While, if taken onboard, these suggestions would likely require substantial revisions to the Methods and Results sections, I do not believe it would require revisiting the actual data or overall narrative.

My primary concern with the paper is that the statistical analysis is rather convoluted and confusingly presented. Specifically, the number of different approaches used to answer the same questions was hard to follow the reasoning behind – as far as I can tell, all of these approaches except for the GLMMs are redundant. I present some detailed feedback on this section below, which I hope will help the authors streamline this.

It is unclear to me why both simple correlation tests and GLMMs were used to test the same questions in this paper. The former approach is seriously confounded by pseudo-replication, since it does not control for the fact that individuals contributed multiple data points (both overall and within classes of gesture-type), and possesses no advantages that I am aware of. The same seems to apply to the ‘compression analyses’ (which I confess not to know much about). The GLMMs deal with these issues, presenting a more robust examination of the data. I would therefore suggest simply dropping the correlation tests and focusing on reporting the GLMMs. If these tests are in fact somehow necessary or justified, then the authors need to explain in detail why this is the case (being the method of choice in previous studies is not sufficient).

However, the reporting of the GLMMs is also generally rather confusing and requires a careful proof-read and rewrite throughout.

For instance, in the examination of Menzerath’s law (**Line 206** on): The authors refer to ‘Model 3’ frequently – however, this is the first time this term has been used and it is not explained until the Methods section at the end of the paper. Even having read the methods section first in my case, the specifics of its usage in the text are not immediately clear. Based on the output of Table 3a the term appears to have been referring to the ‘full’ model for this analysis, which includes every parameter and an interaction – this could be made clearer. However, further confusion comes from the heading of Table 3b– “Model 3 ranking based on AIC values” – there are 5 different models in this table, they are not all Model 3.

Also puzzling is the fact that the authors report that ‘model 3’ fits the data significantly better than the null model (**Line 218**), and state that Size has a significant effect on Duration. This is then immediately contradicted by discussion of the AIC and BIC tables in **Tables 3b** and **3c** respectively, where the full model was in fact the worst fitting model in each table. Indeed, in **Table 3c** it is the null model which appears to have the best fit for the data. The significant difference they refer to comes from an ANOVA comparison, which is irrelevant when more nuanced information criterion methods (AIC + BIC) are also in play. I would suggest removing the ANOVA comparisons entirely and focussing on the latter approach. The authors should also add the standard error of the AIC and BIC to each table, as this allows the reader to interpret whether a difference in IC between two models is notable.

The authors consistently use the terminology of significance testing for their model outputs (Table 2a & Table 3a), but no p-values are reported. Furthermore, what size are the confidence intervals cited? ‘Upper and lower’ is not informative.

Line 159: From reading the text I was uncertain at how exactly Proportion was fit as a predictor in these analyses of Zipf’s Law, and having now checked the raw data and R scripts provided by the authors, I am afraid there is a potentially serious issue here. It is a little difficult to express, so I hope this is clear enough:

The issue lies in how ‘Proportion’ has been defined. It is not that using proportion data is incorrect – e.g. you could use “proportion of plate empty” as a predictor for how much a restaurant patron enjoyed their meal. But the data here is structured quite differently:

- Each row (a recording of a gesture) in the dataset contains a value for ‘Proportion’
- But each ‘proportion’ value is *itself derived from the overall dataset*. i.e. $P \text{ of Gesture } X = (N \text{ gestures} / N \text{ Gesture } X)$.

- This means every gesture of the same type has an identical value for ‘Proportion’: i.e. Every row for Gesture X has $P = 0.47$.

Sequence	Sequence	Gesture	P	Duration	Categ
5	1	Object shake	0.473214	2.88	Manu
6	1	Stomping object/gr	0.005357	2.56	Manu
7	2	Hit object/ground	0.082143	0.36	Manu
7	2	Object shake	0.473214	1.48	Manu
8	2	Object shake	0.473214	0.44	Manu
8	2	Object shake	0.473214	1.84	Manu
9	1	Object shake	0.473214	4.08	Manu
10	1	Object shake	0.473214	4.16	Manu
11	1	Object shake	0.473214	3.68	Manu
12	1	Object shake	0.473214	2.28	Manu
13	2	Object shake	0.473214	3.88	Manu
13	2	Object shake	0.473214	8.96	Manu
14	1	Object shake	0.473214	4.52	Manu
15	1	Object shake	0.473214	5.28	Manu
16	1	Object shake	0.473214	1.24	Manu
18	1	Object shake	0.473214	3.92	Manu
19	1	Object shake	0.473214	1.2	Manu
20	1	Object shake	0.473214	12.72	Manu
21	2	Object shake	0.473214	3.8	Manu
21	2	Stomp object/grou	0.042857	0.2	Manu
22	3	Hit object/ground	0.082143	1.52	Manu
22	3	Object shake	0.473214	3.64	Manu
22	3	Object shake	0.473214	0.72	Manu
23	1	Object shake	0.473214	5.8	Manu
25	1	Object shake	0.473214	1.8	Manu

So, in short: there is only actually one data-point per gesture-type (the overall proportion), but the model treats every row as a new, independent datapoint, resulting in a strange form of pseudo-replication. I hope I have described this issue clearly enough to be helpful.

I am confident there are established ways of resolving this issue, but unfortunately cannot provide them myself.

More clarification points on the analysis:

Line 129: I’m curious what gesture had a duration of 40ms – for context: this is a single frame of video, and roughly the duration it takes for a human eye to move from one fixation point to another.

Line 140: How were outliers qualified?

Line 134: The sample sizes for gestures of 4+ units are extremely small. Given that gesture durations have an extremely large range, from 0.04 to 15.04 seconds (350x longer than the minimum), how representative are these datapoints likely to be? This is not so much of a problem for the Zipf’s law analysis, but raises concerns over examination of Menzerath’s law.

Figure 1A: Drawing a line of best fit through this empty space seems uninformative and potentially misleading.

Line 139: I do not think there is a good reason to abbreviate ‘duration’ and ‘frequency’ to their first letter anywhere in this manuscript (e.g. **Figure 1**). More confusing yet is that **Figure 2** abbreviates duration to ‘t’ instead of ‘d’. Just use the full word throughout.

Line 142: What is ‘L’?

Line 154: ‘whiskers indicate s.e.m.’ – spell out this acronym. I would prefer use of standard deviation for error bars here, as it better represents variability within the data (unclear what SE adds in this particular case). I would also advocate plotting the actual raw data here instead of just the overall means, so that readers can see the distribution of data for themselves. If the authors do insist on using averages, then it seems this would be better served by a boxplot than the current format.

Line 198: Why were only correlation, and not GLMMs used for this section?

Line 219: “When controlling for signaller ID” – not clear what is meant here, all of the models include ID as a random effect.

Line 386: This justification for a one-tailed test seems like circular logic to me. Compression predicts a particular relationship – but this analysis is testing for the existence of compression in this system, so this directionality is not a given, and a relationship in the opposite direction would be just as interesting. In this case it does not seem to matter since the result was non-significant, but I do not think there is a good case for not using a two-tailed test here.

As discussed above, it’s unclear why the correlation or ‘compression’ analysis are necessary at all with the presence of GLMMs below. The latter address the same questions, are more robust, and are not confounded by pseudo-replication.

Line 412: Perhaps there is a good reason for not doing so, but I would think that adding sequence-ID as a random effect would also make sense for the analysis of Zipf’s law. E.g. if I ‘wave’ at someone and then give a ‘thumbs-up’ as part of the same signal, these are not independent data-points (as opposed to a wave and a thumbs-up produced on different days). This could potentially help drive down the noise in the analysis somewhat.

Minor comments

Below, I list some minor suggestions I collected during my readthroughs. Some of these may overlap or be repetitious of comments above.

Line 30 – both of these studies are on chimpanzees, so why generalise to ‘ape’? Seems like an overstatement.

Line 36: ‘signallers consider signalling efficiency more broadly’ – the word ‘consider’ here implies some kind of conscious strategy on the part of signallers, which is not inline with my understanding of these laws.

Line 40: Stylistic point: Quantitative linguistics has identified many statistical regularities across languages. Here the authors should already narrow this field to the specific topic of the paper.

Line 69: As far as I understood the data, Watson et al. identified a likely instance of the menzerath-altman law, but not menzerath’s law (ultimately concluding ‘mixed evidence’).

Line 71: “Chimpanzee gestural communication represents a powerful model in which to explore compression and language laws. Apes have large repertoires of over 70 distinct gesture...”

The authors make several jumps between talking about chimpanzees specifically, and apes more generally, and I get the impression that the two are sometimes conflated.

Line 103: Please clarify the meaning of ‘urgent’ in this context.

Lines 108-110: This needs unpacking a bit – why does being subject to strong selection pressures make them a good model for exploring compression?

Line 116: “following previous studies and to allow for a robust assessment, we also compute compression values related to the respective patterns (Heesen et al., 2019)”

The meaning of this sentence is not clear to me. What are the ‘respective patterns’ being referred to here?

Line 119: Minor phrasing issue - The position of ‘novel’ here makes it sound like the context is novel, rather than the analysis.

Line 125: Do only male chimpanzees produce sexual solicitation gestures? If so, this should be described in the introduction to justify the exclusion of females here (females are not mentioned until Line 313 of the Discussion in the current draft).

Line 123: Is a ‘token’ different from a ‘gesture’? The terms are used quite interchangeably as far as I can tell, so I suggest dropping one.

Line 296 + 297: Using the term efficiency in these two very different ways is not strictly helpful for the reader. I would suggest finding more precise terms (latter use here is more like ‘effective’, not efficient, which is an optimisation between expenditure and effectiveness)

Line 311: “Signallers likely consider signalling efficiency more broadly” – ‘consideration’ of signalling efficiency seems like quite a grand claim about the cognition at work. It’s unclear if this is the claim the authors actually wish to make (in which case I think reference to literature on volitional control of signal structure would be helpful in supporting it), or if it was an accidental implication of this word choice (in which case something more neutral would be appropriate).

Appendix B

Associate Editor Comments to Author

Dear Dr. Hobaiter,

my sincere apologies for taking so awfully long with responding to your submission. We had troubles finding reviewers, have lost two who had agreed along the way, but now can provide three very helpful reviews of your manuscript. All reviewers agreed that substantial work is needed before a final recommendation can be made - and it is possible that such recommendation may be negative. Comments by reviewer 1 on the fundamental definition of units and sequences challenge the idea that the linguistic laws can be applied to the gestural communication investigated here and these concerns are echoed in comments by reviewer 3 on the urgency of gestures, the role of intentionality, and the resulting direction in the two laws studied. All reviewers have issues with the statistical approach and the interpretation of the data that culminate in reviewer 2's suggestion to better view this as a detailed case study with supplemental information. If you feel that you can resolve the definition issues and clearly show how gestures and vocalizations are conceptually and biologically similar enough to warrant application of linguistic laws and answer to the comments on statistic analyses, we will consider a resubmitted manuscript with more carefully phrased conclusions.

Dear Dr Schülke, thank you very much for your work and effort in securing these reviews, they were extremely detailed and constructive, and very helpful in improving the manuscript. We appreciate the opportunity to submit this thoroughly revised version, in which we hope to have now addressed these points and that we hope you will find acceptable for publication.

We have leaned into the case study approach as recommended, we provide a matched analysis of our highly prolific individual (Duane) and the whole dataset, as in doing so we are able to provide some indication of the generalizability of our findings. With the updated, and more nuanced analyses that follow the reviewers' suggestions, we found no difference in the pattern of Duane's data as that in the dataset as a whole – which we hope helps to suggest that our findings are robust (given our small sample).

Reviewer: 1

This is a potentially interesting paper that is clearly written. However, I have serious concerns about the analyses that seem to limit the reliability of the conclusions. My major concern is that it is not clear that units and sequences are defined the same way here as they are in vocal communication. While I appreciate the attempt to expand the application of linguistic patterns in new directions, I am not sure that what is presented here is really comparable. The main problem is that many of the gestures have quite long durations (more than 10 seconds) which does not seem at all comparable to units of speech (syllables or words). In looking at the gestures with long durations it appears that they are all ones that are likely to be repeated. For example, "object shake" can last for anywhere from <1 to 15 seconds. Surely that variation is in not in the speed of a single shaking movement but rather is in the number of iterations of that movement. But repetitions of a movement are themselves a sequence. I understand that there might not be breaks in between (the movement in continuous) but I think the unit of analysis should be the smallest movement element that is repeated (a single movement in one direction or a cycle of back and forth?) and longer examples are considered multiple repetitions of that (so a sequence of X repeats each of short duration). Otherwise, I don't think it is really comparable to vocal communication. There are very few words that repeat the same syllable, and almost never more than once or twice. Bird song can be fairly continuous and last for several seconds but the unit of analysis for questions of compression would be notes or syllables within that song, not the entire song.

Thank you very much for your review of our paper, we write a point-by-point response below. The issue of what makes a unit or a sequence is not a straightforward one indeed. The field itself now extends well beyond the study of vocalizations, including 'units' of analysis that are very varied in form (and length). We can explore the expression of these linguistic 'laws' at different levels, and their study now extends from genomes, and proteins, to non-vocal behaviour such as dolphin tail slapping (Ferrer-i-Cancho & Lusseau, 2009). Even within vocalizations, units of analysis vary – in speech we can consider a spoken word (e.g. Strauss et al., 2007), a syllable (e.g. Rujevic et al., 2021), or a phoneme (Martindale et al. 1996). In signed languages they have been studied at both the level of signs and finger spelling (Börstell et al. 2016).

Gestures, including human gesture, can be considered as a single 'unit' (something akin to a word) and also parsed into smaller sections – for example: the use of the preparation, action stroke, hold-

repetition, and recovery phases (e.g. Kendon, 2004). To take the example of an 'object shake' the smooth repetition of the action stroke (shaking) is here akin to the holding of a 'reach' gesture that is extended and held in place while waiting for a response – both require the continued investment of energy into the production of the gesture unit. It may be possible to invest energy into a gesture for longer than is typically invested into a vocalization – which is limited by breath. Although some primate long calls are also much longer as a single continuous 'unit' than any human speech units tend to be. As is established in gestural research we use a pause, recovery, slowing down or other change in rhythm to separate consecutively produced units of the same gesture type.

As a result, we would respectfully disagree that the gesture 'units' we have defined are not a suitable unit of measurement. But to address this concern, we have now substantially expanded our introduction, providing a richer description of how these laws have been explored in non-vocal units, including in other gestural work, and in our methods we provide more detailed explanation of how we define and distinguish gestural units. We have also expanded on this in the discussion to suggest future comparisons within gestural research that would build on this case-study.

Additionally, treating object shake (and the other repeated gestures) as single point in figure 1 is a little misleading. There were not >250 instances of a gesture that lasted about 2 seconds--there were a few instances of object shake across a wide range of durations (a range that is greater than the range across the means of all call types). The same problem applies to all of the analyses using average durations for gesture types--they are lumping together very different versions of the gestures, some with many repetitions, some with few. This is particularly problematic given that the repetitions themselves might be subject to the laws: e.g., are the shorter versions of object shake more common (Zipf's law, object shake was seemingly more common at shorter durations in figure S5) or are they more likely to appear in longer sequences with other gestures (Manzereth's law)? I know similar linguistic analyses use measures of average duration but the amount of variation within a type is much lower and not due to a variable number of repeats.

Thank you, we very much agree that the issue of lumping gestural (and other unit) instances is indeed a problem, as the field has often relied on correlation-type analyses to explore these relationships. In our original ms we provided both the correlation analyses (that include this problem) and a GLMM approach (in which we have more freedom to control for possible confounding variables).

To address this concern, we now run our GLMM on individual data points, and control for gesture type. We also focus on the GLMM analyses (moving the correlations to Supp Mat – we would like to retain them in the paper as we feel that they provide an important point of like-with-like comparison with much of the previous research in this area, but agree that they should not be a focus). We also now shifted to use Bayesian GLMMs, as these are better suited to smaller datasets.

Another problem relating to the length of some of the gestures is the definition of the end point of the gesture. The second criteria is a change in position if the gesture involves a particular body alignment. This highlights another way that gestures are different from vocal production. Vocal production has to be actively produced--if you stop forcing air over the vocal cords, the sound stops. But a gesture that relies on a certain body alignment is entirely passive (once the posture is achieved). So it is unlikely that such gestures would be subject to the same mechanical pressures/costs. The time an animal is actively moving to produce the gesture is probably more directly comparable to vocal communication. Similarly, the last criteria seems to have more to do with the response of other's than the production of a signal. Response latency seems unrelated to compression and the mechanisms of signal production (response is more relevant to the function of the signal). Given these definitions, it does not seem that duration is entirely comparable across gestures and vocalizations, even for gestures that do not repeat.

While it is true that some gestures include 'hold' positions (Heesen et al., 2019; Hobaiter & Byrne, 2011a), it is not the case that these are 'passive' or without mechanical cost. Whole body postures typically involve holding the body in a stiff or non-relaxed position that requires energetic input to maintain. We would suggest that, at times, the energetic cost involved in maintaining even a 'simple' arm reach gesture extended for several seconds, could be greater than that required to produce a soft vocalization.

As we mentioned above, we fully agree that there may be different parts of a gesture subject to different selective pressures – but also that the exploration of these laws at one level of analysis does not preclude their exploration at others. Moreover, while vocalization provides a rich source of comparison it is not the only point of comparison, with Zipf's and Menzerath's laws expressed across a highly diverse set of systems that encode information from proteins to cities populations.

To address these concerns, we have now provided a richer description of how these laws have been explored in non-vocal units in the introduction, including in other gestural work. In our methods we provide more detailed explanation of how we define and distinguish gestural units, and in our discussion, we describe both the need to consider a range of levels of analysis in gesture, as in vocalization, and also to consider measuring aspects of these signals outside of duration (as is starting to be done in vocalization (e.g. amplitude, Demartsev et al. 2019))

It seems that the Menzerath finding (Figure 2, the compression test, the model in Table 3A) is not given much weight in the manuscript. Figure 2 looks like a pretty clear pattern (up to sequence size of 4 after which there are only 9 data points) but it is largely influenced by a single individual who has the most gestures in the data set (Duane). Taking Duane out of the data set removes the pattern—but it also, by definition, greatly reduces the sample size (by about ½ it appears). Is this just a matter of a smaller sample failing to reach significance despite similar effect size? Is there a reason to remove Duane other than the fact that he had a lot of gestures? I guess I don't see any reason to think that removing Duane makes the data set better and not worse. Duane is described as an outlier but having the most data is not what it means to be an outlier. Also the significant model already contained ID as a random effect (Table 3A). The data set without Duane is given most of the weight in the abstract and discussion but I don't see a good justification for why. I understand removing him and showing that he has a strong influence on the result but I think that the finding based on the entire data set should get priority (in the absence of another reason to exclude Duane).

Thank you for this suggestion; we have now rewritten the manuscript (as suggested elsewhere) to take more of a case-study approach. Doing so has allowed us – as you suggest – to centre the whole dataset, rather than the results without Duane in them. Instead of treating Duane as an outlier we now provide a matched analysis of our whole sample and of Duane's data alone – we find no major differences in the expression of Zipf's or Menzerath's law in either dataset, and we hope that this helps to show that our findings are relatively robust and generalizable.

There appear to be a few inconsistencies in the figures. There are 5 whole body gestures in figure S5 but 6 in figure 1 (and the supplementary Table). In figure 1 it looks like 3 gestures occurred only 1 time but in S5 it looks like 4 did. In S5 some gestures (e.g., Beckon) appear to have a single point that does not match the median line. Thank you for highlighting this, following the updated analyses we have removed Figure 1 (that showed the pattern of duration of gesture types). We have updated Figure S5 (now included in the main ms – Figure 1) and corrected the supplementary Table (which had an additional gesture type listed in error).

Figure S5 is very useful, and I appreciate the inclusion of the raw data. One suggestion for improving these kinds of figures is to arrange the X-axis in some meaningful way (rather than alphabetically) so patterns can be seen more quickly. For example, they could be arranged from highest to lowest median duration.

Thank you very much for this suggestion – we have now re-ordered the boxplots to follow the median values for gesture types.

Reviewer: 2

Comments to the Author(s) (see also attachment 'comments.pdf'):

This is an interesting study that explores the possible existence of two linguistic laws – Zipf's law of brevity and Menzerath's law – in chimpanzee sexual solicitation gestural displays. It is great that the authors use this paper to build upon an earlier study on play gestures in the same study population. Building up a resource of

longitudinal data like this is incredibly useful to the field of animal communication, and specifically, to the growing interest in applying linguistic laws to non-human behavior. My concerns are mostly related to the strong claims made in the manuscript despite there being major limitations in the dataset. This study is really more of a 'case study' that is focused on a small number of gestures from a single animal. I am still very enthusiastic about this study, and I'm sure that collecting these data was no easy task. However, I do want to see some significant changes made to the paper so that it does a better job at representing the dataset and makes more conservative interpretations.

Thank you very much for your positive evaluation of our paper and for your helpful suggestions – we detail how we have incorporated these below. We have, as suggested, reframed the paper as a case-study. Rather than focusing on Duane's data alone (a full case-study approach), we provide a matched analysis of our whole sample and of Duane's data – we find no major differences in the expression of Zipf's or Menzerath's law in either dataset, and we hope that this helps to show that our findings are relatively robust and generalizable.

1. Generally speaking, the emphasis on null results comes across too strong given the study limitations. The results are based on a dataset that is 25% as big as the play gesture study that is referenced throughout the paper, there are only 12 gesture sequences over length 3, and the variation in sequence length is small (n= 6 gestures, while the play study had up to ~16). The null results regarding Zipf's law of Brevity are somewhat convincing, but I suspect that null results involving menzerath law are an outcome of type II error. Figure 2 and the compression test analysis suggest that the law does apply. Also, over 50% of the dataset is from a single individual, which means that this study is more a case study than one that reflects population-level patterns. A dataset that is highly skewed towards one individual isn't very useful for assessing individual variation.

As you suggest, we have substantially re-written the paper to reframe it as more of a case-study and focus on the results from the data including/focused on the individual who contributed most of the data, Duane.

I have three suggestions here. (a) First, the limitations of the dataset should be made more transparent in the Methods/Results and Discussion, including how the dataset differs from the one on play gestures.

Thank you – we have now included much more detail on the limitations of the dataset, for example in the Methods (e.g., Lines 451-455 & 469-473, TC 788-792 & 807-810), and Discussion (e.g., Lines 241-246, TC 463-469); including how it differs from the one on play (e.g., Lines 143-145 & 154-157, TC 196-198 & 207-211).

(b) Second, throughout the manuscript, there should be a toning down of claims that the study showed that the laws "failed" (especially Menzerath law). There are many places where it is said that the laws "failed," and this phrasing is contentious.

Thank you – we had considered that we were being cautious by emphasizing the null results across individuals; however, we understand that given the context of a field where a negative result is unusual, and the limitations of our dataset, that this is a stronger conclusion that we intended. To address this concern, we have substantially adjusted how we describe the null findings – avoiding strong claims of failure (e.g., Line 243, TC 510).

(c) Third, I recommend that the authors lean in to the "case study" aspect of their dataset rather than focusing on individual variation. In Figures 1 and 2, it would be helpful to include subplots that show raw data from the chimp with the majority of datapoints (Duane). In the main text or supplementary results, I'd like to see correlation/GLMM analyses only for Duane. In the Results and Discussion, there should be less emphasis on interpretations related to signaller effects, since the data are not distributed evenly across individuals.

Thank you – we do now lean into the case study aspect as suggested. We do this by contrasting the results from both the whole dataset and Duane as a subset (rather than our previous approach of centring the discussion of the results when excluding Duane). As suggested, we provide subplots that show raw data, and data from Duane alone.

2. Related to comment 1 above, it would help to show raw data so that readers can critique the statistical models and conclusions. I found it difficult to assess the model results because I did not understand the data

structure. Given that the dataset is small, it should be easy to show it in figures. At a minimum, please show raw data from the animal with the most data in the figures on Zipf's law of Brevity and Menzerath's law. It would also help to add a descriptive figure that comes before the Zipf law figure. This figure could show what these gestures are and how they are combined together (e.g., cartoons of the gestures, repertoires of solo vs. sequence gestures, comparison of duration across gesture types and gesture positions with a sequence (1st, 2nd 3rd, etc)). If the authors need space, then move some tables to the supplement. For example, I found it confusing to have tables for BOTH AIC and BIC. Emphasize the most relevant method and move the other to the supplement?

Our initial goal was to provide a range of analyses to facilitate comparison with previous approaches (which are quite varied!); however, we realise this ended up rather messy and distracting. We now take a simpler approach – with the updated GLMMs the focus of the results section, and all non-essential results either cut or moved to the Supplemental. We also now show raw data on the figures – and indicate whether this comes from Duane. While we have not included illustrations of the gesture types here, we do now link in Table S8 (where we describe the repertoire in detail) to our research site (www.greatapedictionary.com) where illustrations and video examples are available.

3. I would like more rationale for why the authors chose the model formulas that they did. First, I do not understand the rationale for why gestures are split by “manual” vs “whole-body” (category in Model 1, PWB in Model 3) other than merely copying the methods from an earlier paper. Based on the supplemental information, it doesn't look like gestures in these categories differ much (i.e., one category isn't longer in duration or more stereotyped than the other). For Model 1, why not use “gesture type” (all 26 types) instead of a 2-level “category”? For Model 3, why not use gesture duration as the dependent variable (instead of average duration), and then include gesture type, sequence position, and sequence id as predictor/random effects? For both models 1 and 3, it seems odd that modified data measures (e.g., average duration, proportions) are used in GLMM analyses instead of the raw data variables. Why not use gesture frequency instead of proportions in Model 1? Why not use gesture token duration instead of averages? Why use PWB in model 3?

Thank you for these suggestions, we have now focused our results section on the models and re-structured them so that

1) we provide a clearer justification for the inclusion of manual vs whole-body (as well as the previous study on Zipf's law in play, this is a common distinction made in gestural research and may be important for comparison across a broader range of findings, including those from signed languages etc. Methods: Lines 439-444, TC 715-723). However, we only include this as a control (to exclude its possible effect) and not as a predictor (as we are not directly interested in what effect it might have)

2) as suggested, we now conduct analyses on raw data from gesture duration/frequency, with gesture type (n=26) included as a random effect.

4. The introduction and discussion need more integration with recent perspectives on linguistic laws and what they mean. I recommend that the authors connect their study to discussions in a in press review in TREE by Semple, Ferrer-i-Cancho and Gustison. This review synthesizes human and non-human literature, discusses what “universality” means, and links these laws to energy expenditure. There are also interesting discussions of these laws and evolutionary processes in papers by Torre and colleagues.

Thank you for directing us to these papers – we have now incorporated a wider range of up to date perspectives in the introduction and discussion as suggested (e.g. lines 104-114, 262-265 – TC 112-122, 529-532).

5. Related to the point above, I recommend that the authors more thoroughly discuss and reference recent human-focused work, since humans are a very relevant comparison to chimps. For example, any discussions of sign language and linguistic laws can be expanded (e.g., L51-110). Potential questions to answer include: Are both Zipf's law of brevity AND menzerath's law tested in human sign language studies? What were the conditions when these laws were tested in humans and are they at all analogous to the present study? (E.g., was sign language studied during naturalistic social interactions?). What different evolutionary pressures may

be at play for human sign language vs. sexual solicitation gestures in chimps?

Thank you for this suggestion, we have now expanded on this in our introduction and also included a more detailed discussion of these points in the introduction and discussion (lines 81-87, 315-321; TC 89-95, 582-588).

Minor comments

- Suggest rephrasing of the paper's title. "Linguistic laws are not the law" is awkward. Only two linguistic laws are tested in this paper, but there are more than two linguistic laws. Also, there are only convincing results to argue against one of these – the Zipf's law of brevity.

We have now changed the title to "Variable expression of linguistic laws in ape gesture: a case study from chimpanzee sexual solicitation."

- L34: Suggest rephrasing of "pressure for efficiency that has been previously proposed to be universal". This is "universality" argument is more nuanced than is suggested in the paper. This point is related to comments above that the authors should integrate more recent perspectives on what "universality" means (i.e., Semple et al. 2021).

Thank you, now rephrased.

- L36: Suggest rephrasing of "highlight that signallers consider signalling efficiency broadly". This phrasing suggests that the chimpanzees are being intentional in how they choose gestures. This conclusion is not very convincing given this study does not get at intentionality. While I agree with the authors that intentionality is an intriguing possibility, it seems like other biological pressures related to arousal states or developmental constraints are more likely.

Thank you, we have now rephrased.

- L36-37: Unclear what is meant by "diverse factors". Suggest being more specific here.

Thank you for highlighting this, we have now changed to "...factors, such as the immediate socio-ecological context of the social interaction." (Line 38-39, TC 45-47).

- L44-45: Rephrase "argued to be present across systems of biological information". This argument is more nuanced than is presented in this manuscript. Again, I recommend the authors refer to the new TREE review by Semple and colleagues.

Thank you, now rephrased.

- L47: Rephrase "is an advantage to choosing the outcome". This sounds intentional, and the study

Thank you, now rephrased.

- L81 (and throughout the manuscript): General use of the word "failure" comes across as contentious. I recommend that the authors tone this down. Typically, the authors are referring to instances when a study was unable to refute the null hypothesis (no correlation); a null result is less convincing than instances when a correlation is found in the opposing direction to a hypothesis.

Thank you, now rephrased throughout.

- L88: Possible typo here, is Zipf law of "abbreviation" meant to be "brevity"?

Thank you, now corrected.

- L103: Could use to develop this rationale a bit more. Why does it matter if sexual solicitations are urgent? Are they actually that urgent? What about energetic costs, what about these solicitations make them costly for males? Why does urgency matter?

- L109: Be more specific here about strong selection pressures. What kind of selection pressures? Is there evidence of female choice for more elaborate signals?

We have now provided a more detailed explanation of the solicitation context for chimpanzees, however, our undefined use of the term urgent was unhelpful sorry. It stems from early gestural work based in captivity, where the majority of gestures were used in play and were described as

being 'less-evolutionarily urgent' (Tomsello & Call, 2007; and placed in contrast to primate communication in the wild that was related to alarm calling or food etc.). We now provide a more detailed explanation of the context.

- L111-19: Could use more background information here about what is known about the composition of sexual solicitation gestures. How long are they? Do they vary within and across individuals? How do females respond to them (e.g., what parts of these gestures might be sensitive to selection pressures)?

Unfortunately, there is relatively already published about gestural behaviour in this context, we do now provide more detail – largely taken from Hobaiter & Byrne, 2012 – but we also take the opportunity to describe their composition and length etc. in more detail in the descriptives at the start of our results section.

- L119, L302, and L328: Suggest rephrasing of “novel evolutionarily urgent context” and “evolutionary salient scenarios”. It is not clear what these phrases actually mean. Does this mean that evolution happens quickly in sexual contexts?

Terms now removed, as suggested, in the rewrite of this section.

- L125: Use median instead of mean since there is high skew in the dataset.

Thank you, now implemented throughout the result section.

- L127-132: How many sequences per male?

This information now added (line 193-195, TC 294: Sequences ranged from 1 to 181 per male (median:8±44.54))

- L141: Could use a sentence here describing what a “compression test” is. Not immediately obvious what this means.

Thank you, now added (line 465-471 – TC 804-810: which looks at whether the expected mean code length observed in the dataset is significantly smaller than a range of mean code lengths calculated via permutations).

- L142-144 (and throughout the manuscript): It was not clear to me at all what was meant in the terminology related to the compression tests. How is a gesture “big” or “small”? What does this mean? It is explained a bit in the Methods section, but could use a more general interpretation in the results section.

Thank you, in the Supplementary material we have now included a richer description of what is meant by ‘big’ and ‘small’.

- L176-178: If BIC is more parsimonious, then why not just stick with that approach instead of including both? Personally, I find the inclusion of both AIC and BIC more confusing than helpful.

Thank you, given the updated analysis, model ranking was excluded from the manuscript. We compute full-null model comparisons using the Level One Out information criterion.

- L201-202: Why are the correlation test p-values different for the manual gestures across these two sentences? There should be one p-value for the positive correlation? If the first analysis is a one-tailed test and the second analysis is a two-tailed test, then please make this clear.

Thank you, we have now specified it in the supplementary material containing the correlation analysis (see Supporting Information Lines 94-95).

- L208 (and throughout): Rather than say “there was no relationship”, say “we found no evidence for a relationship”. Technically, correlation/GLMM tests are used to refute a null hypothesis, but they are not appropriate to “prove” that a null hypothesis exists. Also, on line 209, the correlation test ($p=0.076$) suggests that there is a trend in the hypothesized direction. Why isn't this trend acknowledged? Marginal effects are acknowledged elsewhere in the paper (L205).

Thank you for highlighting this inconsistency; now corrected (Supporting Information Lines 82, 98, 134).

- Tables: The table titles could use more descriptive titles than “model 1” and “model 3”. How about including whether the models are testing Zipf’s law of brevity or menzerath’s law?

Table headings and model names now edited to be more informative.

- Where is model 2? The results and Tables appear to jump from 1 to 3. I assume that I missed some fine print stating that model 2 is in the supplement? This is confusing.

Thank you, we have now re-named the models to Zipf-model and Menzerath-model so that their names are more useful to the reader.

- L216-217: I did not understand this phrase “a linear association between n and t that could not be sufficiently captured by the Spearman correlation test.” Could use more explanation here. I thought that spearman tests don’t assume that data are linear?

Thank you for highlighting this, we have now removed the terms in the rewrite of the manuscript.

- L261: Suggest using more conservative phrasing instead of “... did not follow either Menzerath’s law or Zipf’s law of brevity”. This is a strong conclusion given that the dataset is limited and that correlation tests are not designed to prove a null hypothesis. It is strange to me that the authors say that there is no evidence for Menzerath’s law when Figure 2 looks quite convincing and the Spearman correlation shows a trend in the hypothesized direction. I suspect that this was an instance of Type II error and not having enough data across sequence sizes and individuals.

Thank you for highlighting this inconsistency. We have updated the analysis using Bayesian GLMMs as these are better suited to smaller datasets. The current analyses showed a pattern coherent with Menzerath’s law with a negative relationship between sequence size and gesture duration, which is also reflected in the figures.

- L263: Unclear what “no subsets” means. Explain.

Thank you, following the analysis update we have now reworded the discussion and removed the terms.

- L270-279: This paragraph would also be a good place to develop reference to human sign language. One of these human studies is cited but is not discussed in any detail in the manuscript.

Thank you for the suggestion, we have now included a paragraph where we take a deeper look at human sign language studies (lines 297-303, TC 564-570).

- L270-279 and L310-328: These are another places where links to Semple et al would be useful. “Universality” should be used in a more nuanced way than is depicted in this paragraph.

Now included.

- L306+: The authors should consider adding in an experimental human study (Kanwal...Kirby 2017 Cognition) to this part of the discussion centered on “urgency” and in situ contextual variation. This human study tests the presence of zipf law of brevity when people are faced with pressures to produce efficient vs. accurate communication signals. Both pressures were needed for individuals to communicate in ways that supported Zipf’s law of abbreviation.

Thank you for highlighting this study. We have now taken it into account when rewriting the discussion (lines 328-331, TC 596-598).

- L331-346: There needs to be more detail about when and how sexual solicitation gestures were recorded. What efforts were made to insure that males were observed as evenly as possible (e.g., focal observations, similar times of the day, etc)? What recording equipment was used? Where males always being recorded on video during focal samples? If video samples were only taken during sexual solicitations, then how often were gestures not caught on film because they occurred before the filming started?

Filming natural behaviour of wild chimpanzees in a visually dense rainforest is not straightforward, and we employ fairly strict criteria for inclusion that inevitably limited our sample size. However, we

felt that it was important to be cautious – particularly where there was any doubt about sequence length. We now provide more detail on these aspects in our methods (e.g. lines 362-371, TC 633-642).

How many gesture sequences were there and what was the range in sample sizes across males? How many females were involved in these recorded interactions, and were the same females involved across all males? We now provide these details in the start of our Results section. While, given our sample size, it was not possible to control for recipient ID in our analyses we do now highlight that it is important to consider more factors like this in future work (lines 181-202, 343; TC 269-302; 612).

- L359: This “1s” threshold to separate sequences is somewhat arbitrary. Can the authors provide a histogram of inter-gesture interval lengths to show that this threshold is appropriate for their study system?

Unfortunately, the duration of inter-gesture intervals was not recorded for this study. We used the 1s minimum following (Heesen et al., 2019) and other gesture studies in which this is a fairly well-established interval for distinguishing sequence types (e.g., Hobaiter & Byrne, 2011b).

- L429-441: The compression test approach could use more explanation. I still cannot understand it, especially what it means to be “big” or “small” (is this the same this as short or long duration?). It might help to show the observed parameters vs. permuted data as a panel in the relevant Results figures.

Thank you, we have now included a richer description in the supplementary material. We have also included plots in the supplemental showing the observed vs. the permuted data.

Reviewer: 3

Comments to the Author(s) (see also 'RSOS Review 061221.pdf' attached):

The paper is well-written and I enjoyed reading it. The data provides useful insights into chimpanzee gestural communication and the growing literature on the applicability of linguistic laws to animal communication. I think it will make a valuable addition to this literature. However, there are a handful of issues which should be addressed before publication. I am confident these can mostly be addressed by providing additional details and argumentation, or streamlining the content. I have packaged these comments into the attached .pdf file for formatting reasons.

Thank you very much for this positive assessment of our paper and for your helpful comments and suggestions – we have addressed these point by point below.

Firstly, there are a few conceptual points which I believe the manuscript would greatly benefit from having unpacked a bit further by the authors:

Conceptual issues

The paper often refers to the fact that sexual solicitation gestures are ‘urgent’, and I think this needs to be unpacked a bit more. Solicitation gestures are ‘urgent’ in the sense that unsuccessful solicitation means you are somewhat less likely to have offspring over the course of your lifetime, but this is very different to the kind of temporal urgency inherent to e.g. alarm signals (where things need to happen quickly or else you get eaten). It’s therefore not immediately clear why a gesture with a long duration would impact reproductive success (do chimps always wait until a gesture is finished before responding? Why is a fast response important?), whereas a prolonged long alarm call has obvious potential fitness consequences. Indeed, in birds, for example, more elaborate sexual displays are more successful.

We do now provide a more detailed description of chimpanzee sexual solicitation behaviour in the introduction. Unfortunately, there is very limited study of these signals to date, and so there is no available work we know of at present that looks at variation in their success with changes in signalling. However, while we agree that there is less ‘urgency’ than in a predator alarm call, we highlight this here because gesture has historically been described as being of particularly low urgency (based on early work which was conducted in captivity and had concluded that gesture was

primarily used for communication in play). We highlight the important of sexual solicitations to individual fitness, but have toned down/removed language around its 'urgency'.

*The authors touch on this at points, hinting that fast gestures may be easier to conceal from dominant rivals. However, this is not factored into their analysis (e.g. fitting rank as a predictor, since dominant individuals should not experience pressure towards compression), and does not sit well with the fact that these gestures often seem to occur in sequences (making them more salient again). My understanding of the literature is that gestural sequences in chimps are primarily persistence and elaboration on the initial signal (used as evidence of intentionality) – my prediction would then be that gestures later in a sequence might have a longer duration than those that come earlier, as they become increasingly exaggerated to elicit the intended behavioural response. E.g. **Line 303**: “the costs of signal failure are high, ” – this makes absolute sense for signals such as alarm calls where the costs of not being heard are direct, and the benefits of redundancy are clear. However, it is not clear precisely how this applies to sexual solicitation gestures, and needs to be unpacked by the authors. E.g. what is failure constitutes in this context, and why redundancy would mitigate it. If failure is simply not having your signal recognised, then there is near-zero cost (except the very minor energetic cost of producing the signal) – you can simply try again. If failure is a prospective mate seeing your signal and turning down copulation, it is unclear how redundancy would mitigate this.*

This is our fault for not providing a detailed explanation in the methods which touched only very briefly on our definition of a sequence. Gestural sequences take on two forms: one is persistence (including elaboration) following failure of earlier signals; however, others are combined rapidly, with less than 1-second between signals and no behavioural evidence of response waiting (e.g. checking of the recipient's behaviour). Here the signaller produces the sequence independently of recipient behaviour. Early work on the function of gesture 'sequences' often conflated these two types – but they show different patterns of use in wild chimpanzees (Hobaiter & Byrne 2011). In this study we only consider the second type: gestural sequences produced with less than 1-second between signals and that did not meet criteria for response-waiting.

We agree that subsequent gesturing following failure may lead to the exaggeration or extension of gesture tokens to elicit a response; however, these cases are excluded by our criteria, which instead focus on sequences that were produced independently of, rather than in response to, recipient behaviour.

To address this point, we now provide a more detailed explanation to highlight this important distinction (lines 393-403; TC 667-678).

Comments on the Statistical analysis

While my comments on this section are quite extensive, I wish to emphasise that I do not think the analysis has necessarily been carried out incorrectly. Indeed, I broadly agree with the authors' interpretation of the data. My comments and suggestions largely concern eliminating redundancy and improving clarity to improve the reader experience. While, if taken onboard, these suggestions would likely require substantial revisions to the Methods and Results sections, I do not believe it would require revisiting the actual data or overall narrative.

My primary concern with the paper is that the statistical analysis is rather convoluted and confusingly presented. Specifically, the number of different approaches used to answer the same questions was hard to follow the reasoning behind – as far as I can tell, all of these approaches except for the GLMMs are redundant. I present some detailed feedback on this section below, which I hope will help the authors streamline this. Thank you very much for these detailed suggestions – we have been able to address these statistical issues in our new ms using updated (and now Bayesian) GLMMs and moving other analyses (such as the correlations) to the Supplement (as these largely serve to provide like with like comparisons to earlier research).

It is unclear to me why both simple correlation tests and GLMMs were used to test the same questions in this paper. The former approach is seriously confounded by pseudo-replication, since it does not control for the fact that individuals contributed multiple data points (both overall and within classes of gesture-type), and

possesses no advantages that I am aware of. The same seems to apply to the ‘compression analyses’ (which I confess not to know much about). The GLMMs deal with these issues, presenting a more robust examination of the data. I would therefore suggest simply dropping the correlation tests and focusing on reporting the GLMMs. If these tests are in fact somehow necessary or justified, then the authors need to explain in detail why this is the case (being the method of choice in previous studies is not sufficient).

Thank you, we agree that the correlation tests are problematic – but as they were widely used across papers in this field, we had included them in the original ms to ease like-with-like comparison. As suggested, we now exclude these from the main paper, moving them to the Supplement, and we highlight there the issues with pseudo-replication, advising caution in their interpretation.

However, the reporting of the GLMMs is also generally rather confusing and requires a careful proof-read and rewrite throughout. For instance, in the examination of Menzerath’s law (**Line 206** on): The authors refer to ‘Model 3’ frequently – however, this is the first time this term has been used and it is not explained until the Methods section at the end of the paper. Even having read the methods section first in my case, the specifics of its usage in the text are not immediately clear. Based on the output of Table 3a the term appears to have been referring to the ‘full’ model for this analysis, which includes every parameter and an interaction – this could be made clearer. However, further confusion comes from the heading of Table 3b– “Model 3 ranking based on AIC values” – there are 5 different models in this table, they are not all Model 3.

Following suggestions here and from the other reviewers, we have now fundamentally restructured our updated results section so that it is much simpler in format, and we provide meaningful names for our models to help the reader.

Also puzzling is the fact that the authors report that ‘model 3’ fits the data significantly better than the null model (**Line 218**), and state that Size has a significant effect on Duration. This is then immediately contradicted by discussion of the AIC and BIC tables in **Tables 3b** and **3c** respectively, where the full model was in fact the worst fitting model in each table. Indeed, in **Table 3c** it is the null model which appears to have the best fit for the data. The significant difference they refer to comes from an ANOVA comparison, which is irrelevant when more nuanced information criterion methods (AIC + BIC) are also in play. I would suggest removing the ANOVA comparisons entirely and focussing on the latter approach. The authors should also add the standard error of the AIC and BIC to each table, as this allows the reader to interpret whether a difference in IC between two models is notable.

Thank you, we have now run a new Bayesian analysis only including one predictor. We have performed model comparison using the Level One Out (LOO) information criterion between the full model and the null model without the predictor.

The authors consistently use the terminology of significance testing for their model outputs (Table 2a & Table 3a), but no *p*-values are reported. Furthermore, what size are the confidence intervals cited? ‘Upper and lower’ is not informative.

Thank you, we have run a new analysis using Bayesian models, and report the 95% credible intervals in tables S4, S5, S6, S7 in Supporting Information - section 3.

Line 159: From reading the text I was uncertain at how exactly Proportion was fit as a predictor in these analyses of Zipf’s Law, and having now checked the raw data and R scripts provided by the authors, I am afraid there is a potentially serious issue here. It is a little difficult to express, so I hope this is clear enough: The issue lies in how ‘Proportion’ has been defined. It is not that using proportion data is incorrect – e.g. you could use “proportion of plate empty” as a predictor for how much a restaurant patron enjoyed their meal. But the data here is structured quite differently:

- - Each row (a recording of a gesture) in the dataset contains a value for ‘Proportion’
- - But each ‘proportion’ value is itself derived from the overall dataset. i.e. P of Gesture $X = (N \text{ gestures} / N \text{ Gesture } X)$.

- This means every gesture of the same type has an identical value for ‘Proportion’: i.e. Every row for Gesture X has $P = 0.47$.

So, in short: there is only actually one data-point per gesture-type (the overall proportion), but the model treats every row as a new, independent datapoint, resulting in a strange form of pseudo-replication. I hope I have described this issue clearly enough to be helpful.

I am confident there are established ways of resolving this issue, but unfortunately cannot provide them myself.
Thank you for highlighting this. To address it we now use the raw data (each individual gesture instance) rather than the average for each type. We have also included gesture type as a random variable, which informs the model of the dependency between the gesture instances that belong to the same type (as previously it would have considered them independent).

More clarification points on the analysis:

Line 129: I'm curious what gesture had a duration of 40ms – for context: this is a single frame of video, and roughly the duration it takes for a human eye to move from one fixation point to another.

Thank you for your interest, two gestures had a gesture of 0.4s (minimum unit). Both were 'hit object/ground' and were performed in sequences with other gestures, thus the 'preparation' 'hold' and 'recovery' phases of the gesture performance were essentially eliminated as the gesture action followed directly on from a previous gesture and moved directly into a subsequent one.

Line 140: How were outliers qualified?

We no longer exclude outliers in the updated analyses.

Line 134: The sample sizes for gestures of 4+ units are extremely small. Given that gesture durations have an extremely large range, from 0.04 to 15.04 seconds (350x longer than the minimum), how representative are these datapoints likely to be? This is not so much of a problem for the Zipf's law analysis, but raises concerns over examination of Menzerath's law.

Thank you for bringing this up to our attention. We have now included raw datapoints in the Menzerath's law figures (Figure 2, 3) which show how many data we have for each sequence size. We have decided to keep them in the analysis but have highlighted in the discussion that these should be treated with caution (lines 253-259, TC 520-526).

Figure 1A: Drawing a line of best fit through this empty space seems uninformative and potentially misleading. Given the updated results we have now excluded the figure from the manuscript.

Line 139: I do not think there is a good reason to abbreviate 'duration' and 'frequency' to their first letter anywhere in this manuscript (e.g. **Figure 1**). More confusing yet is that **Figure 2** abbreviates duration to 't' instead of 'd'. Just use the full word throughout.

Thank you, now corrected to full word throughout, as suggested.

Line 142: What is 'L'?

Thank you for highlighting the need for clarification. "L" represents the expected mean code length of a gesture type in the sample. It is calculated via equation 1 in supplementary material 1, it is then compared to the distribution of all possible L's created by permutating the data $R=10^5$ times. It is considered to be 'small' if it falls within the lower 5% of the distribution.

Line 154: 'whiskers indicate s.e.m.' – spell out this acronym. I would prefer use of standard deviation for error bars here, as it better represents variability within the data (unclear what SE adds in this particular case). I would also advocate plotting the actual raw data here instead of just the overall means, so that readers can see the distribution of data for themselves. If the authors do insist on using averages, then it seems this would be better served by a boxplot than the current format.

Thank you very much for this suggestion, we now include boxplots with jittered points for the Menzerath's law figures, which show the raw datapoints.

Line 198: Why were only correlation, and not GLMMs used for this section?

Thank you, in the updated analysis and results we have moved the correlation analysis to the supplementary so that the focus is on the GLMMs (but the correlation remains available for comparison with previous work as this approach has been widely used in earlier work).

Line 219: "When controlling for signaller ID" – not clear what is meant here, all of the models include

ID as a random effect. As discussed above, it's unclear why the correlation or 'compression' analysis are necessary at all with the presence of GLMMs below. The latter address the same questions, are more robust, and are not confounded by pseudo-replication.

As suggested, we have now removed the correlation and compression tests from the main Results section.

Line 412: Perhaps there is a good reason for not doing so, but I would think that adding sequence-ID as a random effect would also make sense for the analysis of Zipf's law. E.g. if I 'wave' at someone and then give a 'thumbs-up' as part of the same signal, these are not independent data-points (as opposed to a wave and a thumbs-up produced on different days). This could potentially help drive down the noise in the analysis somewhat.

Thank you for the suggestion, we have now included Sequence ID as a random effect in both models.

Minor comments

Below, I list some minor suggestions I collected during my readthroughs. Some of these may overlap or be repetitious of comments above.

Line 30 – both of these studies are on chimpanzees, so why generalise to 'ape'? Seems like an overstatement. Now corrected to chimpanzee.

Line 386: This justification for a one-tailed test seems like circular logic to me. Compression predicts a particular relationship – but this analysis is testing for the existence of compression in this system, so this directionality is not a given, and a relationship in the opposite direction would be just as interesting. In this case it does not seem to matter since the result was non-significant, but I do not think there is a good case for not using a two-tailed test here.

Thank you, with the correlation analysis we were specifically testing for a negative correlation reflective of the presence of a pattern following Zipf's law. Following such precise hypothesis and prediction we decided to perform one-tailed tests. We also report two-tailed results for comparisons.

Line 36: 'signallers consider signalling efficiency more broadly' – the word 'consider' here implies some kind of conscious strategy on the part of signallers, which is not inline with my understanding of these laws.

Now edited to "the expression of ape gestures appears shaped by other factors..." (line 38, TC 45).

Line 40: Stylistic point: Quantitative linguistics has identified many statistical regularities across languages. Here the authors should already narrow this field to the specific topic of the paper.

Thank you, now cut in updated ms.

Line 69: As far as I understood the data, Watson et al. identified a likely instance of the menzerath-altman law, but not menzerath's law (ultimately concluding 'mixed evidence').

Thank you, Menzerath-Altman's law is a power-law-exponential formula based on the more general Menzerath's law (here analysed). Menzerath-Altman's law mathematically predicts the expected length of an element based on the number of its constituents (Semple et al., 2022).

Line 71: "Chimpanzee gestural communication represents a powerful model in which to explore compression and language laws. Apes have large repertoires of over 70 distinct gesture..." The authors make several jumps between talking about chimpanzees specifically, and apes more generally, and I get the impression that the two are sometimes conflated.

Thank you for pointing this out – we have now checked over the ms to tidy up our usage.

Line 103: Please clarify the meaning of 'urgent' in this context.

We have clarified our use of 'urgent', and also cut/edited its use in this section.

Lines 108-110: *This needs unpacking a bit – why does being subject to strong selection pressures make them a good model for exploring compression?*

We now do so – please see lines 133-162 (TC 188-217) in the introduction.

Line 116: *“following previous studies and to allow for a robust assessment, we also compute compression values related to the respective patterns (Heesen et al., 2019)” The meaning of this sentence is not clear to me. What are the ‘respective patterns’ being referred to here?*

Now cut in the updated ms.

Line 119: *Minor phrasing issue - The position of ‘novel’ here makes it sound like the context is novel, rather than the analysis.*

Now adjusted to make it clear that it is the analysis that is novel.

Line 125: *Do only male chimpanzees produce sexual solicitation gestures? If so, this should be described in the introduction to justify the exclusion of females here (females are not mentioned until Line 313 of the Discussion in the current draft).*

We now mention that these are present, but were excluded (as they are fairly infrequent, and typically not in sequences in this population; lines 378-380, TC 649-651).

Line 123: *Is a ‘token’ different from a ‘gesture’? The terms are used quite interchangeably as far as I can tell, so I suggest dropping one.*

We’re sorry, this is an example of where the methods after ‘main paper’ approach is unhelpful. A token is a specific linguistic term, we use it to discriminate from gesture types. We now provide a definition in lines 183, as well as in the methods in 384-392 (TC 271, 656-666) and have tidied up our usage to hopefully avoid ambiguity.

Line 296 + 297: *Using the term efficiency in these two very different ways is not strictly helpful for the reader. I would suggest finding more precise terms (latter use here is more like ‘effective’, not efficient, which is an optimisation between expenditure and effectiveness)*

Thank you, we have now tidied up our usage throughout.

Line 311: *“Signallers likely consider signalling efficiency more broadly” – ‘consideration’ of signalling efficiency seems like quite a grand claim about the cognition at work. It’s unclear if this is the claim the authors actually wish to make (in which case I think reference to literature on volitional control of signal structure would be helpful in supporting it), or if it was an accidental implication of this word choice (in which case something more neutral would be appropriate).*

Our phrasing was at fault here – now reworded to a more neutral framing (line 323, TC591).

Appendix C

Dear Dr. Hobaiteer,

Your original reviewers were kind enough to provide a second round of comments. In view of these comments I suggest major revisions to address the issues raised including additional analyses and plots. I am confident these revisions will significantly add to the accessibility of your research and look forward to receiving a new version of your contribution.

Best wishes,
Oliver Schülke

Dear Dr Schülke, thank you very much for your and the reviewers constructive feedback, it has helped us to substantially improve the manuscript. We have now addressed the concerns in a revision of the text, in new analyses requested, and in a point-by-point response below. For example, we have now moved some information from the discussion to the introduction to clearly set up gesture structure, as we all now providing the analyses of data excluding Duane (in the supplemental, to focus on the case study approach) and additional data to control for R1's concerns about the structure of different gesture types. Our findings are unchanged, but are now hopefully more clearly supported and presented.

Best wishes, Cat Hobaiteer and Alexandra Safryghin

Reviewer: 1
Comments to the Author(s)

I appreciate the extensive revisions and I think the manuscript is improved. I have two lingering concerns. First, while the focus on the gestures from Duane is improved, I am not sure it is handled in the best way. To me, the question is, do the gestures of Duane show a different pattern than the gestures from everyone else? To answer that, you would want to run the analyses on the other (non-Duane) gestures alone. Analyzing all the gestures together and separating out Duane does not answer this (all gestures together may show the pattern because it is being driven by the large number of Duane samples—the reader can not tell). I can piece the story together across the revisions (because, if I recall correctly, in the previous version the non-Duane samples did not show the pattern) but it should be explored here (with the new analysis methods). What would it mean to have the pattern show up in one sample but not the other? I think it is ok to have the story be somewhat complicated.

Thank you very much for your feedback. While we didn't want to detract from the new 'case study' approach in the revised ms, we understand the importance of not entirely cutting out the non-Duane data. To address this, we now included the analysis on non-Duane data in the supplementary information 3 as well as including a figure depicting the pattern of Menzerath's law for gestures performed by each individual (Supporting information 6).

We refer to these results briefly in the main ms and direct readers to the supporting information for full details (e.g. lines 218; 245/TC248; 278). In the data excluding Duane we found no effect of Zipf's law, and no effect of Menzerath's law. However, the individual plots suggest sample size may be an issue in detecting these patterns. For example, we see a clear pattern in Duane's data (273 gestures), and an increasingly less clear pattern for each individual as the sample gets smaller (for example in the next most prolific individual with n=72 gestures the pattern appears visible for the first three sequence sizes, but this falls off rapidly in the individuals with smaller samples (supporting information 6).

My second issue is one I raised previously—that repeated (and held postural gestures) may be different from other units analyzed in sequences. Previously I had pointed out that they are not like

vocal sequences—and I agree with the response that it makes sense to look at these laws in all kinds of systems, not just vocal sequences. However, repeated (and held gestures) are not like the other gestures that they are being compared to within this manuscript (some of which seem to be constrained to a short duration). I did not make this point clear in my previous review but this could create problems beyond the theoretical issue of mixing sequences (in the case of repeated gestures) and units in the same analysis. For example, it could be that the ‘prone to long duration’ gestures (repeated or held) are rarely embedded in longer sequences. This would produce the Menzereth pattern but not because of ‘compression’, rather because of different uses of different gesture types. This seems relatively easy to test for—is there any evidence of Menzereth occurring within the common gesture types? Does separating the ‘prone to long duration’ call types change things? Thank you very much for this comment. After a thorough look at the data, we can demonstrate that the Menzereth’s law pattern is not driven by a decrease in the use of those you call ‘prone-to-long’ (or loose) duration gestures in longer sequences.

We would like to clarify that loose duration gestures represent the majority of gesture types (and gesture tokens) employed by the chimpanzees ($n=20$ out of 26 types; $n=456$ loose duration tokens out of 560 tokens); thus, ape gesturing is largely characterised by use of these ‘loose’ gestures.

To demonstrate that our findings are not due to a difference in the use of these gestures, in the supplemental material and in the main text we now

1. Provide a more detailed explanation of loose vs fixed gesture types in the introduction – here we moved a paragraph from the discussion (where’d we’d touched on gesture structure) into the introduction to help facilitate this early on for the reader (lines 128-150/TC137-160), we also now provide more detail in the methods (lines: 482-497/TC537-547).
2. We provide a graph showing the distribution of the frequency of sequences formed of only loose duration gestures, only ‘common’ gesture types (or fixed duration) and a mix of both (supporting information 4, Figure S4). The graph shows how even longer sequences of gestures still contain loose gestures, either in sequence with other loose gestures or together with fixed duration gestures. On the other hand, the use of sequences made of only fixed duration gestures is quite rare ($n=23$ sequences of length 1 and $n=6$ sequences of length 2) and sequences of only fixed duration gestures reach a maximum length of 2 tokens.
3. We also provide a graph showing the distribution of the duration of gestures in sequences that are only made of these fixed duration gestures (supp information 4, Figure S5). Here, the duration decreases when comparing their use in sequences of length 1 and length 2. These results show how even the fixed duration gestures still show compression when put together in a sequence.

We describe these findings in the ms (lines 268-272/TC301-304).

Reviewer: 2

Comments to the Author(s)

The authors did a great job at addressing my feedback. This version of the manuscript is streamlined, and most of the results are communicated clearly. At this stage, I only have comments related to the Zipf law analysis and the Figures. I am confused by how Zipf’s law was tested in this version of the manuscript. In addition, there is little visual representation of the data analyses, which makes it hard to evaluate statistical output and interpretations. These comments should be straightforward to address.

1. Zipf Law analysis: Zipf’s law of brevity is simply defined as a negative association between unit frequency and unit duration. In this dataset, units are gesture types ($n=26$), and the (mean or median) duration and frequency of occurrence for each of these 26 gestures should be plotted against each other. It doesn’t seem like the revised analysis approach directly tests Zipf’s law of brevity, and it’s unclear to me what is actually being tested. Based on the S4 and S5 tables, it looks

like Proportions (for all gestures per all sequences?) were used as a predictor variable, and Gesture ID was a random effect variable. Therefore, there are 560 (or 290) datapoints, instead of 26. To directly test Zipf's law of brevity, I recommend using a Bayesian model with 26 data points, where gesture type duration is predicted by gesture type frequency, with gesture category (whole body, manual) included as another predictor variable or as a random effect.

Thank you very much for your comment. We have now included the suggested analysis in the supporting information 5, as well as including a comment in the main manuscript (lines 241-242/TC273-274). The full model and the null model were similar, so the effect of frequency on the distribution of the median duration of the 26 gesture types was negligible. Thus, we continue to find no evidence supporting a Zipf's law pattern in these data. One drawback of this type of analysis is the impossibility of controlling for individual effects, which may result in an inaccurate and unreliable analysis, so we restricted full details of this model to the supp mat.

2. Figure 1: There doesn't seem to be a visual representation of the analysis related to Zipf law. I recommend that the authors add a subplot or two to Figure 1 to show what the data for these analyses (Table S4 and S5) looked like. I suspect this plot would look something like Figure 2-3, where there is a full dataset and a Duane-only subset (see comments for Figure 2-3). Something similar to Figure 1 from the initial submission of the manuscript would be great.

Thank you very much for this comment. We have now included two graphs in the main text showing the distribution of gesture types and their frequency of use for all individuals and for Duane only (Figure 2).

3. Figure 2-3: Figure 3 is essentially a repetition of Figure 2. I recommend that the authors combine these figures since they are the same dataset and analysis approach.

Thank you, we have now combined the graphs in one pane.

4. In all Figures, please include results of statistical tests where appropriate inside the figures themselves. When using a Bayesian model approach, the regression line (slope and intercept from the model) can be overlaid on top of the data. In addition, the significance level of the test can be included on the figure.

Thank you, we have now plotted the regression slope over the figures.

5. In all Figures, please define the axes and units of measurement in the legends. For example, in Figures 2-3, it is not clear what "sequence size" means and what the points represent (each point = gesture? Or each point = sequence?).

Thank you, we have now updated the description to: "Points represent individual gesture tokens, ordered by the length of the sequence they were performed in".

6. Additional supplemental figure: It would be helpful to provide a couple figures showing individual animal data related to Zipf-model and Menzerath-model (e.g., 16-panel figures similar to Figure S8 in initial submission). Providing such figures would allow readers to understand the individual variation in the data, which has become an important issue in some recent papers.

Thank you, we have now included the requested graph in supporting information 6.

Reviewer: 3

Comments to the Author(s)

I commend the authors for the hard work they have put into their substantial revisions on this manuscript. They have dealt with my major conceptual issues with the original manuscript, and the statistical analysis is now much easier to follow. I have just two concerns about this revised analysis which I describe below:

1) The authors now acknowledge that the great majority of their data comes from a single individual ('Duane') and have changed their analysis as a result: Analysing all of the individuals together, and Duane separately. The authors claim that doing so validates the generalisability of their findings, but I am not sure what this means.

"As our data may be particularly influenced by a single prolific individual (Duane) who contributed around half of the data, we assess the generalizability of our findings by replicating analyses conducted on the full dataset on a subset of the data containing only gestures by Duane."

How exactly pooling Duane's data with the other individuals tells us about the generalisability of these findings is not clear at all in terms of statistical inference. The authors need to unpack their argumentation much more on this matter.

"As a result we consider it a case-study; nevertheless, our findings were similar for both the full dataset across male signallers, and for a single prolific individual, as well as in a range of alternative analyses (Supporting Information 1), suggesting that the pattern of results appears to be relatively robust."

I am not convinced. When there is an effect present in only Duane's data, the fact that a weaker effect is also found in a dataset where he contributes over half of the datapoints, is unsurprising. To convincingly show that the effect is not entirely driven by Duane, the authors would surely need to demonstrate that the effect is present in a dataset that does -not- include Duane's data?

Thank you for your comment. We have now included the analysis of non-Duane data in the supporting information 3 (and describe the findings briefly in the main ms; e.g. lines 218; 245/TC248; 278). We now acknowledge more clearly that the pattern is largely driven by Duane's data but also describe how the distribution of the datapoints for the next most-prolific individuals show a tendency towards a Menzerath's law pattern, suggesting that there may be issues in our ability to detect the pattern in smaller datasets (lines 263-265//TC296-298).

2) I am also concerned that Duane's data was not analysed appropriately when examined individually.

Line 452: "For the models testing Duane's data, signaller ID was removed from the random factors."

A GLMM using 260 datapoints from a single individual, without random effects, will treat this as 1 datapoint from each of 260 individuals (repeating the pseudoreplication issue I highlighted in my previous review), falsely increasing the 'power' of the model. Including a random effect with one level is also rather meaningless, so I think an entirely different approach is needed to analyse Duane's data effectively. Unfortunately, I am not practiced enough in analysing $N = 1$ data to recommend an alternative, so expert statistical advice should be sought.

Having said all this, the fact that the relationship holds when Duane's data is pooled with the others and a random effect of ID is introduced does alleviate my concerns somewhat.

Thank you for your suggestion. We have re-run the analysis for Duane, and now include the date data was collected on as a random factor. By including date (which has multiple levels), we address concerns about pseudoreplication and demonstrating that our data are representative of Duane as an individual, and not biased towards data collected on a particularly prolific day and are representative of his behaviour as an individual.

Minor comments

Line 33: I suggest replacing "fail to find" with "did not find", as the former implies not being able to find something that is actually there, or that we at least wanted to be there.

Thank you for your suggestion – changed to 'did not find'.

Lines 116-117: These sentences do not really make sense to me. i) one explanation for what?

Thank you for your comment, we have now changed it to 'One explanation for a repertoire-level absence of Zipf's law of brevity'.

ii) *what case of long-distance calls?*

Thank you, we have now updated the text to 'as seen in some long-distance signals'.

iii) *"impacted its expression" is very vague,*

Thank you, we have now changed to: "is that the context in which signals are produced may impact the emergence of such patterns".

iv) *"in this case" which case?*

Thank you for your remark, now changed to: "Specifically, in the case of chimpanzee gestures, the absence of a pattern resembling Zipf's law of brevity may result from the analysed context: play."

Line 136 & 278: As in my previous review, I urge the authors to nail down the meaning of the term "urgent" or else not use it at all.

Thank you, now changed to: biologically 'relevant'.

*Line 156: Unclear how a solicitation gesture might lead to lethal aggression from a neighbouring group. Do the authors mean when attempting solicitation with a female *from* a neighbouring group?*

Thank you, now added a sentence (line 184/TC 210): 'For example, during consortships individuals may travel to the boundaries of their home area, increasing the risk of encounters with neighbouring individuals.'

Line 164: I find the addition of this methods primer to the Introduction very useful for this format of manuscript. However, the movement between tenses ("we test" \ "we will test" \ "we tested") through the manuscript is not ideal. The tests have already been carried out – it is only their presentation to the audience that has not occurred yet.

Thank you, now updated to past tense throughout.

Line 210: What was the R^2 value? It would be just as important that it is not <1 . Just give the raw value, it should be = 1 in all cases (or else there is an issue with convergence).

Thank you, now changed.

Line 223: " R^2 values <1.02 ."

Same issue as above.

Thank you, now changed.

Line 223: the term "significant effect" is not really appropriate when applying Bayesian methods. Please rephrase throughout. More importantly, the effect reported does not seem to be 'significant' as it has been reported:

" $b = -0.18$, s.d. = 0.04, 95% CrI [0.26, -0.11]; Figure 2)."

Thank you, now changed to 'substantial'.

For a robust effect I would expect to see credible intervals that do not overlap at all with 0. I see from inspecting the supplementary material that '0.26' is simply missing a minus symbol. Please insert. Indeed, there are a number of typos throughout the manuscript that have crept in with the revised text. I encourage the authors to carry out a careful proof-read.

Thank you for your feedback, we have now thoroughly proof-read the manuscript for errors.

Line 336: “while many vocalisations are relatively fixed”

Fixed in what sense? Support with references.

Thank you for your comment. We have now added references: Fitch et al. (2016); Janik and Slater (1997) (lines 371/TC 425).

Line 423: Apologies for not picking up on this in the last draft, but one would ideally expect to see inter-observer reliability testing reported. While it’s important that raters are consistent with themselves, a consistently incorrect rater is also of no use.

We apologise but after several weeks of digging into this to try to address it we are unable to offer inter-observer reliability.

We no longer have the file that links the the original dataset to the exact video file name and time at which each gesture marked was located – unfortunately even using date+gesture type proved impossible as there were many gestures of the same type recorded on the same day between the same individuals. We realise that this is an issue and we are frankly embarrassed by our inability to address this. In our more recent coding schemes there are clear identifiers for every gesture coded and inter-observer measures are the norm, but as this was an older dataset these are not available. The only way to address this would have been to recode the entire dataset and run inter-observer analysis on the new data – months of work and to some extent an entirely new dataset. The only other step we could take would be to provide measures of the reliability of the main coder (AS), who is an extremely experienced gesture coder, on their coding of the same measures in other gesture datasets. But we realise this is also not entirely satisfactory. Instead to provide transparency and highlight this limitation, we have added a statement in the methods explaining that these data were only checked by intra-observer testing. We hope that as intra-observer (or indeed no inter-observer measures) have been the norm in other datasets used in this field we are at least no less robust than has been typical, but we fully realise the importance of being clear about this shortfall. (Line 463-464/TC 517-518).

Line 446-7: “Random factors” – random intercepts, random slopes, or both? Given the focus of the analysis on determining whether Duane is an outlier or representative of the general population, I imagine random slopes would be appropriate to allow for individual variation.

Thank you for your comment. Unfortunately, because the sample sizes from many individuals outside of Duane are quite small, we do not have sufficient gesture types across individuals, or sequences of each size across individuals to address this properly. As a result, we were unable to include interactions between fixed effects and random slopes because we have incomplete combination matrices.